# CONTENT-AWARE MAMBA FOR LEARNED IMAGE COMPRESSION

**Yunuo Chen**[1]  **Zezheng Lyu**[2]  **Bing He**[1]  **Hongwei Hu**[3]  **Qi Wang**[3]
**Yuan Tian**[4]  **Li Song**[1]  **Wenjun Zhang**[1]  **Guo Lu**[1*]
[1]Shanghai Jiao Tong University   [2]Massachusetts Institute of Technology
[3]Alibaba Group   [4]Shanghai AI Laboratory

## ABSTRACT

Recent learned image compression (LIC) leverages Mamba-style state-space models (SSMs) for global receptive fields with linear complexity. However, the standard Mamba adopts content-agnostic, predefined raster (or multi-directional) scans under strict causality. This rigidity hinders its ability to effectively eliminate redundancy between tokens that are content-correlated but spatially distant. We introduce Content-Aware Mamba (CAM), an SSM that dynamically adapts its processing to the image content. Specifically, CAM overcomes prior limitations with two novel mechanisms. First, it replaces the rigid scan with a content-adaptive token permutation strategy to prioritize interactions between content-similar tokens regardless of their location. Second, it overcomes the sequential dependency by injecting sample-specific global priors into the state-space model, which effectively mitigates the strict causality without multi-directional scans. These innovations enable CAM to better capture global redundancy while preserving computational efficiency. Our Content-Aware Mamba-based LIC model (CMIC) achieves state-of-the-art rate-distortion performance, surpassing VTM-21.0 by 15.91%, 21.34%, and 17.58% in BD-rate on the Kodak, Tecnick, and CLIC datasets, respectively. Code will be released at `https://github.com/UnoC-727/CMIC`.

## 1 INTRODUCTION

Image compression plays a vital role in computer vision and multimedia by reducing storage and bandwidth while preserving visual quality. Learned image compression (LIC) has advanced rapidly with end-to-end training and differentiable loss functions (Ballé et al., 2017; Ballé et al., 2018; Du et al., 2025; Li et al., 2025b). Early LIC methods used convolutional neural networks (CNNs) for nonlinear analysis and synthesis transforms (Minnen et al., 2018; He et al., 2022). Recently, transformer-based designs expanded receptive fields to capture long-range dependencies (Zou et al., 2022; Zhu et al., 2022; Liu et al., 2023; Li et al., 2024a). However, transformers achieve a global receptive field at the cost of quadratic time complexity. State-space models (Gu & Dao, 2023; Zhu et al., 2024) offer a compelling alternative: they couple global receptive fields with linear complexity and have already been adopted in several LIC models (Qin et al., 2024; Zeng et al., 2025).

Yet, Mamba's Selective Scan mechanism, originally conceived for one-dimensional sequences, faces two fundamental challenges when adapted for image compression. **First, its rigid, content-agnostic scanning order hinders effective redundancy removal.** Vanilla Mamba processes tokens based on their fixed spatial proximity (i.e., raster scan), failing to account for feature correlation and content similarities. Effective compression, however, relies on capturing dependencies between semantically related regions, which may be spatially distant. By separating content-correlated tokens while grouping unrelated ones, the fixed scan path weakens Mamba's ability to model dependencies between tokens that are distant in Euclidean space but close in feature space, leading to suboptimal rate-distortion (RD) performance. **Second, Mamba is a causal sequence model, and the strict causality is misaligned with the non-causal nature of images.** For SSM, when an image is scanned into a sequence, each token is processed based only on the preceding tokens in the raster-scan order. Since a token can only access information from its predecessors, the model ignores the

---
*Corresponding Author

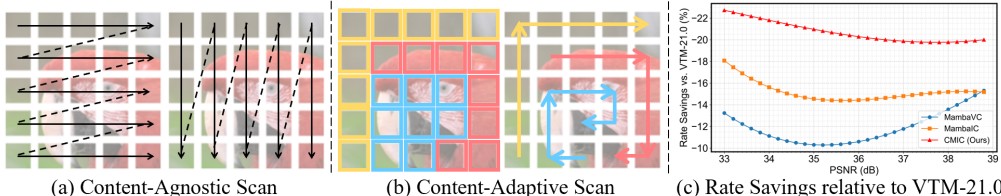

| (a) Content-Agnostic Scan | (b) Content-Adaptive Scan | (c) Rate Savings relative to VTM-21.0 |

Figure 1: (a) Illustration of standard 2D Selective Scan. (b) Illustration of our content-adaptive scan: content-correlated tokens are scanned consecutively to better eliminate redundancy. (c) Rate savings relative to VTM-21.0 on the Tecnick dataset. CMIC significantly outperforms two SOTA Mamba-based LIC models: MambaVC (Qin et al., 2024) and MambaIC (Zeng et al., 2025).

context of subsequent tokens. While multi-directional scanning (Liu et al., 2024a; Guo et al., 2024b) is a common solution, this approach quadruples computational complexity and still relies on the content-agnostic, Euclidean-based paths (Fig. 1). These limitations collectively reduce the model's ability to capture global dependencies, which are critical for high-performance image compression, highlighting the need for a more flexible and adaptive scanning mechanism.

To overcome these limitations and enhance the content-awareness of Mamba, we introduce Content-Aware Mamba, a dynamic SSM designed specifically for compression. First, CAM addresses the content-agnostic fixed scan by prioritizing interactions between content-correlated tokens. It achieves this by clustering latent tokens and reorganizing the token sequence to group similar tokens together. This ensures that tokens are processed based on their feature-space proximity rather than their original spatial arrangement. By allowing the scan to follow paths of high content similarity, we empower Mamba to capture long-range dependencies more effectively while maintaining its linear complexity.

Second, to mitigate Mamba's inherent causality without the cost of multi-directional scanning, we introduce a prompting mechanism. We modulate the state-space representation by injecting sample-specific prompts constructed from clustering results and a distribution-aware dictionary. This prompt encodes global statistics to modulate SSM's hidden states, allowing information from the entire image to influence the sequence modeling process at every step. Consequently, information retention is no longer determined solely by preceding tokens but is also guided by the global priors. This approach effectively mitigates the strict causal chain without quadrupling the computational load.

These two strategies enable efficient, content-aware, and non-causal long-range modeling. Built upon this foundation, our Content-Aware Mamba-based LIC model, CMIC, achieves state-of-the-art (SOTA) RD performance, surpassing the traditional codec VTM-21.0 by 15.91%, 21.34%, and 17.58% in BD-rate on the Kodak, Tecnick, and CLIC datasets. As illustrated in Fig. 1 (c), our model also demonstrates a significant performance gain over recent advanced Mamba-based LIC models.

Our main contributions can be summarized as:

- **Content-Adaptive Token Permutation.** We propose a novel token permutation mechanism that reorders the scan sequence based on feature similarity. This prioritizes feature-space proximity over Euclidean spatial adjacency, significantly strengthening Mamba's ability to capture long-range redundancy.

- **Global-Prior Prompting.** We introduce a redundancy-aware prompt dictionary that conditions the SSM on sample-specific global priors. This approach relaxes Mamba's strict causality and enhances global modeling capabilities with minimal computational overhead.

- **End-to-End CMIC Model.** We build CMIC, a content-aware Mamba-based LIC model that surpasses previous Mamba-based LIC models in both RD performance and efficiency.

## 2 RELATED WORK

### 2.1 LEARNED IMAGE COMPRESSION

End-to-end learned image compression (LIC) has rapidly progressed in recent years. The original CNN-based method by (Ballé et al., 2017) and its extension that couples VAEs with hyper-priors (Ballé et al., 2018) laid the groundwork for LIC. Subsequent studies have mainly advanced RD performance by: stronger analysis–synthesis transforms and more expressive entropy models (Zafari et al., 2023; Mentzer et al., 2022; Ma et al., 2019; Lu et al., 2022; Zhang et al., 2023; Gao et al., 2021; Fu et al., 2023; Bégaint et al., 2020; Li et al., 2025a; Liang et al., 2025; Chen et al., 2025b).

Early CNN variants introduced residual blocks (Cheng et al., 2020) and octave residual modules (Chen et al., 2022; Fu & Liang, 2023), while Xie et al. (2021) explored invertible architectures. INR-based methods have also been explored (Kim et al., 2024; Leguay et al., 2023). Transformer-based architectures have surged in LIC research, including Transformer variants (Zou et al., 2022; Wang & Ling, 2024), hybrid CNN-Transformer schemes (Liu et al., 2023) and frequency-aware window attention mechanisms (Li et al., 2024a). Recently, researchers have employed linear attention models, such as state-space model for the non-linear transform networks to balance the RD performance and model complexity (Qin et al., 2024; Zeng et al., 2025; Wu et al., 2025). Recently, SegPIC (Liu et al., 2024b) uses semantic masks and dynamic CNNs to achieve region-adaptive transforms. Zhang et al. (2024b) employs a clustering scheme to rearrange the feature map, subsequently applying CNNs to the rearranged features. In terms of entropy models, advances include the autoregressive model by Minnen et al. (2018), checkerboard modeling (He et al., 2021), channel-wise models (Minnen & Singh, 2020), and the space-channel context model of He et al. (2022). More works propose quadtree hierarchies (Li et al., 2022b; 2023), Transformer-based entropy models (Qian et al., 2022), and multi-reference schemes (Jiang et al., 2023; Jiang & Wang, 2023). T-CA enhances channel autoregression with transformers (Li et al., 2024a).

## 2.2 STATE-SPACE MODELS

State-space models (SSMs) were first developed in control theory (Kalman, 1960). They are now popular in deep learning as their sequential updates cost only linear time in sequence length, which is ideal for modeling long-range relationships (Gu et al., 2021; Smith et al., 2022; Fu et al., 2022).

The newest breakthrough is Mamba (Gu & Dao, 2023; Dao & Gu, 2024). Researchers have already applied Mamba beyond language to vision tasks (Zhu et al., 2024; Liu et al., 2024a; Yue et al., 2024; Li et al., 2024b). Some works focus on low-level tasks: MambaIR (Guo et al., 2024b) tackles local-pixel forgetting and channel redundancy of Mamba while applying it to single-image super-resolution; FreqMamba (Zhen et al., 2024) operates in the Fourier domain for deraining; MambaLLIE (Weng et al., 2024) modifies the state-space equation for low-light enhancement; and further adaptations address dehazing, deblurring, rain removal, and fusion tasks (Zheng & Wu, 2024; Gao et al., 2024; Wu et al., 2024; Xie et al., 2024; Qiao et al., 2024). Recently, SSMs are also adopted by learned image compression models such as Qin et al. (2024) and Zeng et al. (2025).

## 2.3 CONTENT-ADAPTIVE IMAGE COMPRESSION

Content-adaptive processing is widely recognized as an effective way to improve the complexity-performance trade-off in vision models. For example, Routing Transformer (Roy et al., 2021) and Adaptive Token Dictionary (Zhang et al., 2024a) realize content-adaptive sparse attention, enabling stronger global receptive fields. In learned image compression, adaptivity has also been actively explored. Li et al. (2022a); Wang et al. (2023); Fu et al. (2024a); Jeong et al. (2024) introduce deformable CNNs and Tang et al. (2022); Yang et al. (2021); Spadaro et al. (2024) employ Graph Neural Networks (GNNs) to achieve adaptive compression. Zhang et al. (2024b) further enhance CNN-based LIC with a coarse, grid-anchored clustering scheme.

Our work aims to make Mamba content-adaptive while preserving its linear-time global modeling. In Mamba-based compression, the key issue is not the Transformer-style complexity–globality trade-off, but that content-agnostic, raster-order scanning poorly matches the redundancy structure among tokens. To address this, we introduce a fine-grained, non-Euclidean clustering strategy tailored for Mamba-based compression. More detailed clustering comparisons are provided in Appendix A.2.

## 3 METHOD

### 3.1 PRELIMINARY

Fig. 2 (a) illustrates the pipeline of our content-aware image compression model, which primarily follows a standard VAE layout and comprises non-linear transform networks and an entropy model. The analysis transform $g_a(\cdot\,;\boldsymbol{\theta}_a)$ encodes an input RGB image $\boldsymbol{x}$ into latent features $\boldsymbol{y} = g_a(\boldsymbol{x};\boldsymbol{\theta}_a)$, where $\boldsymbol{\theta}$ refers to the learned parameters. These features are discretized by uniform quantization $Q$, $\hat{\boldsymbol{y}} = Q(\boldsymbol{y} - \boldsymbol{\mu}) + \boldsymbol{\mu}$, where the mean parameter $\boldsymbol{\mu}$ is provided by the entropy model. The synthesis transform $g_s(\cdot\,;\boldsymbol{\theta}_s)$ then reconstructs the image $\hat{\boldsymbol{x}} = g_s(\hat{\boldsymbol{y}};\boldsymbol{\theta}_s)$.

The hyper-encoder $hp_a(\cdot;\boldsymbol{\phi}_a)$ maps $\boldsymbol{y}$ to side information $\boldsymbol{z}$, quantized to $\hat{\boldsymbol{z}}$. The hyper-decoder $hp_s(\cdot;\boldsymbol{\phi}_s)$ predicts mean and scale parameters $(\boldsymbol{\mu},\boldsymbol{\sigma})$; together with the spatial–channel context $\boldsymbol{\varphi}$,

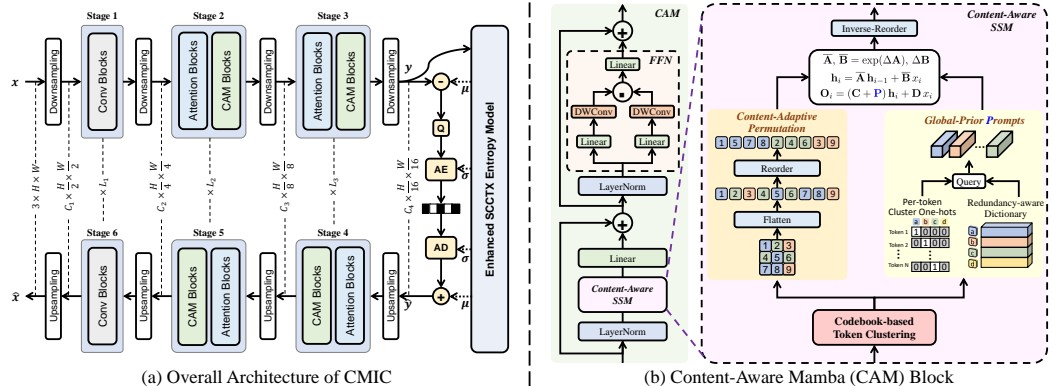

(a) Overall Architecture of CMIC | (b) Content-Aware Mamba (CAM) Block

Figure 2: **Overview of the Proposed Method.** (a) The CMIC framework. Feature dimensions are set as $\{C_1, C_2, C_3, C_4\}$, and the six non-linear transform stages have depths $\{L_1, L_2, L_3, L_3, L_2, L_1\}$. Panel (b) shows the Content-Aware Mamba block, detailing the Content-Aware SSM architecture. Numbers 1-9 represent the indices of tokens, while letters a-d denote distinct cluster categories.

they define a conditional Gaussian $p_{\hat{y}}(\hat{y} \mid \boldsymbol{\mu}, \boldsymbol{\sigma}, \boldsymbol{\varphi})$ used for entropy coding and the bitrate term $R$.

$$\boldsymbol{z} = hp_a(\boldsymbol{y}; \boldsymbol{\phi}_a), \quad \hat{\boldsymbol{z}} = Q(\boldsymbol{z}), \quad \boldsymbol{\mu}, \boldsymbol{\sigma} = hp_s(\hat{\boldsymbol{z}}; \boldsymbol{\phi}_s).$$

$$R = \mathbb{E}\left[-\log_2 p_{\hat{y}}(\hat{y} \mid \mu, \sigma, \varphi)\right] + \mathbb{E}\left[-\log_2 p_{\hat{z}}(\hat{z})\right].$$

The network is optimized via the classic rate-distortion loss $\mathcal{L}_{\text{RD}}$, where $d$ denotes image distortion:

$$\mathcal{L}_{\text{RD}} = R + \lambda\, d(\boldsymbol{x}, \hat{\boldsymbol{x}}),$$

## 3.2 OVERVIEW OF CONTENT-AWARE MAMBA-BASED LIC

First, we give a quick overview of how our model works with Content-Aware Mamba blocks. We split its non-linear transform into six stages based on feature resolution. For each stage, inspired by Liu et al. (2023); Li et al. (2024a), we first utilize window-attention to capture fine-grained local dependencies, while our proposed CAM blocks are introduced to enhance long-range modeling.

As shown in Fig. 2 (b), the core part of CAM block is a Content-Aware SSM. It first clusters tokens into several categories, and permutes the flattened token sequence to bring similar tokens into proximity, enhancing local coherence in the token sequence. Meanwhile, sample-specific prompts are generated from the prompt dictionary. The reordered tokens, infused with these prompts, are processed by a 1-D selective scan for efficient long-range modeling. Our efficient entropy model is built upon the SCCTX model (He et al., 2022). As illustrated in Fig. 3, we incorporate depthwise convolution for context modeling and employ gated MLPs for parameter aggregation. More details and ablations are provided in Appendix A.3. We next detail two key components for Content-Aware SSM: *Content-Adaptive Token Permutation* and *Global-Prior Prompting*.

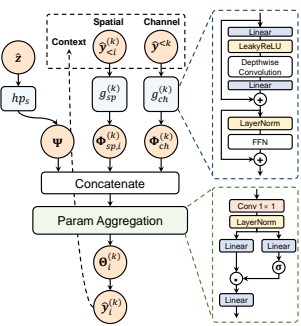

Figure 3: Our Entropy Model

## 3.3 CONTENT-ADAPTIVE TOKEN PERMUTATION

A key limitation of the standard Mamba model for image compression is its fixed raster-scan sequence, which processes tokens in a predetermined spatial raster-scan order. This rigidity is suboptimal for image compression as effective redundancy removal requires interactions between content-similar tokens, regardless of their spatial position. Although clustering provides a direct solution for grouping related tokens, naive implementations like per-sample online K-Means (McQueen, 1967) are impractical. They are not only computationally expensive but also highly unstable for training due to the repeated centroid re-initialization and sensitivity to early-stage features.

**Codebook-based Token Clustering.** To achieve efficient and stable clustering, we draw inspiration from VQ-VAE (Van Den Oord et al., 2017) and instead employ a shared, learnable codebook

containing all the centroids. Analogous to the vector codebook in VQ-VAE, the centroids in the codebook define a discrete latent space that captures semantically meaningful visual features for fine-grained clustering. This shared codebook, updated via an exponential moving average, ensures training stability and robustness to initialization, yielding high-quality clustering results, as evidenced by Fig. 10. Crucially, at inference time, our method requires no iterative updates, resulting in efficient and deterministic assignments.

Specifically, given the feature map $\mathbf{X} \in \mathbb{R}^{H \times W \times d}$, the feature at each position is treated as a token $\mathbf{x}_i$, thus we have $\mathbf{X} = \{\mathbf{x}_i\}_{i=1}^N \subset \mathbb{R}^d$. We employed a cosine-based clustering to group tokens into $K$ clusters $\mathcal{G}_1, \ldots, \mathcal{G}_K$, each with normalized centroids $\mathbf{c}_1, \ldots, \mathbf{c}_K$. To initialize the centroids, the tokens of first batch are divided into $K$ consecutive segments, and each centroid $\mathbf{c}_k$ is computed as the average of the token embeddings within its corresponding segment. Each CAM block holds its own cluster centroids, which are shared across all images and are only updated in the training phase.

**During training**, we update cluster centroids using the K-Means algorithm (Algorithm 1). For $T$ iterations, each token $x_i$ is assigned to its nearest center $c_j$ based on the cosine-derived distance matrix $Distance_{i,j}$, yielding assignments $\{g_i\}$. New centroids $c_j^*$ are then computed as the normalized mean of all tokens assigned to each cluster. Centroids for empty clusters remain unchanged. Finally, the updates are smoothed using an exponential moving average (EMA) with decay $\lambda$.

Each CAM block maintains and updates its own codebook of cluster centroids. This mechanism enables each block to dynamically adapt to its local feature distribution across the whole training dataset, leading to an efficient and precise grouping of content-similar tokens throughout the network. Furthermore, by aggregating statistics across many batches, this non-gradient update process allows the codebook to encapsulate knowledge of the dataset-level feature distribution. More details about this clustering strategy and stability are discussed in Appendix A.8-A.10.

---

**Algorithm 1:** Training-phase Cosine K-Means

**Require:** tokens $\mathbf{X} = \{\mathbf{x}_i\}_1^N$, centroids $\{\mathbf{c}_j\}_1^K$, iterations $T$, decay $\lambda$

1: $\{\mathbf{c}_j^\star\}_1^K \leftarrow \{\mathbf{c}_j\}_1^K$
2: **for** $t \leftarrow 1$ **to** $T$ **do**
3:      $Distance_{i,j} \leftarrow \frac{\mathbf{x}_i^\top \mathbf{c}_j^\star}{\|\mathbf{x}_i\|_2 \|\mathbf{c}_j^\star\|_2}$
4:      $g_i \leftarrow \arg\max_j Distance_{i,j}$
5:      **for** $j = 1$ **to** $K$ **do**
6:          $\mathbf{c}_j^\star \leftarrow \begin{cases} \frac{\sum_{i:g_i=j} \mathbf{x}_i}{\|\sum_{i:g_i=j} \mathbf{x}_i\|_2} & \text{if } \exists\, i : g_i = j \\ \mathbf{c}_j^\star & \text{otherwise} \end{cases}$
7:      **end for**
8: **end for**
9: $\mathbf{c}_j \leftarrow \lambda\, \mathbf{c}_j + (1-\lambda)\, \mathbf{c}_j^\star$
10: **return** $\{g_i\}_1^N, \{\mathbf{c}_j\}_1^K$

---

Given the updated centroids, final assignments $g_i \in \{1, \ldots, K\}$ are determined by the cosine similarity between tokens and cluster centroids. We build a permutation $\pi : \{1, \ldots, N\} \to \{1, \ldots, N\}$ that first groups all tokens with $g_i = 1$, then those with $g_i = 2$, and so on. Applying $\pi$ yields a reordered sequence $\tilde{\mathbf{X}} = \mathbf{X}_{\pi(\cdot)}$, so tokens belonging to the same cluster become contiguous in 1-D sequence, encouraging interactions between similar tokens. This enables SSM to focus on proximity in feature space. The inverse permutation $\pi^{-1}$ is cached and applied to restore the original spatial layout.

**During inference**, input tokens are deterministically and efficiently assigned to clusters using the final updated centroids, without requiring any further K-Means updates.

### 3.4 GLOBAL-PRIOR PROMPTING

The vanilla Mamba layer models token interactions using a discrete state-space system. Its state is computed recurrently according to the following equations:

$$\bar{\mathbf{A}} = \exp(\Delta \mathbf{A}), \quad \bar{\mathbf{B}} = (\Delta \mathbf{A})^{-1}(\exp(\Delta \mathbf{A}) - \mathbf{I}) \cdot \Delta \mathbf{B}.$$
$$\mathbf{h}_i = \bar{\mathbf{A}}\, \mathbf{h}_{i-1} + \bar{\mathbf{B}}\, \mathbf{x}_i, \quad \mathbf{O}_i = \mathbf{C}\, \mathbf{h}_i + \mathbf{D}\, \mathbf{x}_i,$$

where $\mathbf{x}_i, O_i \in \mathbb{R}^d$ are the $i$-th input and output tokens, $\mathbf{h}_i \in \mathbb{R}^{d_s}$ is the hidden state, and the state matrices $\bar{\mathbf{A}}$ and $\bar{\mathbf{B}}$ are discretized from continuous-time parameters. The matrices $\mathbf{C}$ and $\mathbf{D}$ project the hidden state and input back to the output space. The output matrix $\mathbf{C}$ acts analogously to the query in self-attention, determining which information is read from the hidden state (Guo et al., 2024a).

However, the recurrent formulation imposes a strict causal structure. Each token's updated state depends only on its predecessors in the scan sequence, which limits the model's awareness of the global context. Even with our content-adaptive scan, which encourages interactions between similar tokens, this inherent one-directional information flow persists. To mitigate this, we propose modulating the

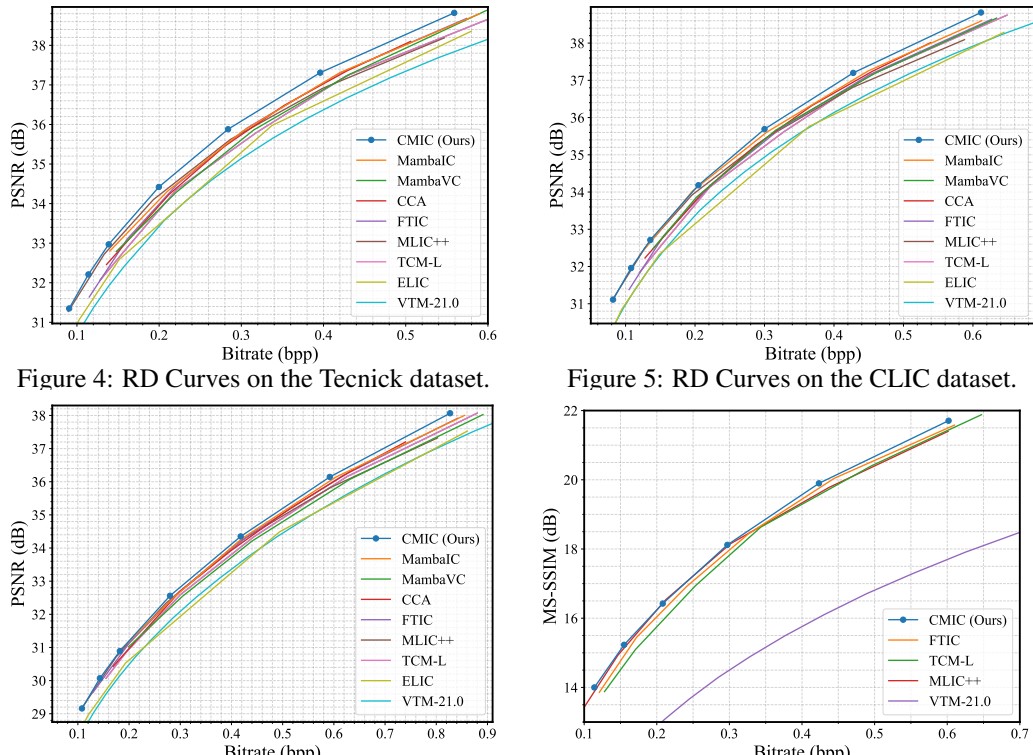

Figure 4: RD Curves on the Tecnick dataset.   Figure 5: RD Curves on the CLIC dataset.

Figure 6: Performance evaluation on the Kodak Dataset.

output matrix $\mathbf{C}$ with global priors. This prior summarizes the overall token distribution of the input, making it more adaptive to the token distribution of each specific instance.

**Redundancy-aware Prompt Dictionary.** To this end, we introduce a distribution-aware dictionary $\mathbf{U} \in \mathbb{R}^{K \times d_s}$ to further exploit the clustering prior, where $K$ is the number of clusters. Each entry in this dictionary is a prompt vector corresponding to a semantic cluster. Specifically, we apply a learnable linear projection $\mathcal{A} : \mathbb{R}^d \to \mathbb{R}^{d_s}$ to each centroid and obtain

$$\mathbf{U} = \mathcal{A}\big([\mathbf{c}_1; \ldots; \mathbf{c}_K]\big) \in \mathbb{R}^{K \times d_s},$$

so that each row of $\mathbf{U}$ serves as the prompt vector associated with a cluster. This differs from MambaIRv2 (Guo et al., 2024a), where the prompt pool is a standalone learnable matrix optimized directly by gradient descent without explicit semantic constraints; in contrast, our prompt dictionary is explicitly tied to the redundancy-aware clustering centroids. For an input with $N$ tokens, we form a one-hot assignment matrix $\mathbf{\Gamma} \in \{0, 1\}^{N \times K}$ representing the cluster membership of each token, as shown in Fig. 2 (b). A sample-specific prompt matrix $\mathbf{P} \in \mathbb{R}^{N \times d_s}$ is then generated by querying the dictionary $\mathbf{P} = \mathbf{\Gamma}\mathbf{U}$. As a result, the prompt signal reflects how redundancy is distributed across semantic clusters, highlighting clusters with higher or lower redundancy for the current features. Crucially, whereas the centroid codebook is updated via non-gradient K-Means, the mapping $\mathcal{A}(\cdot)$ is differentiable and trained end-to-end for the final rate-distortion objective.

**Prompt Conditioning in Mamba.** To enhance global content-awareness, following the Attentive State-Space equation in MambaIRv2 (Guo et al., 2024a), the sample-specific prompt $\mathbf{P}$ is injected into the state-space equation by augmenting the output projection matrix $\mathbf{C}$ as follows:

$$\mathbf{O}_i = (\mathbf{C} + \mathbf{P})\mathbf{h}_i + \mathbf{D}\mathbf{x}_i$$

This mechanism combines global statistical knowledge from the learned dictionary with the sample-specific semantic layout from the clustering. By conditioning the output of SSM on this prompt, the model encodes global priors during the scan process, effectively relaxing the strict causal constraint and enhancing its ability to model long-range dependencies.

## 4 EXPERIMENTS

### 4.1 EXPERIMENTAL SETUP

We train our CMIC models on the Flickr2W dataset (Liu et al., 2020), optimizing with the Adam optimizer (Kingma, 2015) and an initial learning rate of $10^{-4}$. Distortion is quantified using two

| Method | BD-rate (%) | | | Complexity | | | |
|--------|-------------|---------|--------|-------------|---------|-------------|---------------|
| | Kodak | Tecnick | CLIC | Params (M) | TFLOPs | Latency (s) | Peak Mem (GB) |
| VTM-21.0 | 0.00 | 0.00 | 0.00 | – | – | – | – |
| ELIC | −3.10 | −7.41 | −0.84 | 33.29 | 1.74 | 0.335 | 1.50 |
| MLIC++ | −13.42 | −16.73 | −13.94 | 116.48 | 2.64 | 0.547 | 2.08 |
| WeConvene | −6.85 | −10.17 | −7.03 | 105.51 | 4.82 | 1.293 | 4.53 |
| TCM | −10.04 | −10.42 | −8.60 | 75.89 | 3.74 | 0.542 | 7.73 |
| CCA | −11.99 | −13.53 | −11.40 | 64.89 | 3.28 | 0.385 | 5.04 |
| FTIC | −12.94 | −13.89 | −10.21 | 69.78 | 2.38 | >10 | 4.90 |
| S2CFormer | −14.28 | −17.20 | −12.88 | 79.83 | 4.17 | 0.362 | 8.32 |
| HPCM | −14.33 | −16.77 | −14.54 | 68.49 | 2.21 | 0.304 | 2.97 |
| MLICv2 | −16.16 | −20.13 | −15.79 | 84.30 | 2.78 | 0.520 | 5.07 |
| DCAE | −15.40 | −19.62 | −16.46 | 119.22 | 2.28 | 0.478 | 5.59 |
| LALIC | −13.68 | −17.26 | −15.01 | 63.24 | 2.53 | 0.385 | 4.09 |
| MambaVC | −8.10 | −11.82 | −10.94 | 47.88 | 2.10 | 0.425 | 14.73 |
| MambaIC | −13.01 | −15.27 | −15.23 | 157.09 | 5.56 | 0.669 | 20.32 |
| **CMIC (Ours)** | −15.91 | −21.34 | −17.58 | 69.11 | 2.39 | 0.405 | 4.44 |

Table 1: **Quantitative comparisons of SOTA LIC models.** BD-rates are computed over VTM-21.0. FLOPs, peak memory and decoding latency are measured on 2K-resolution images.

metrics: mean square error (MSE) and multiscale structural similarity (MS-SSIM). The Lagrangian multipliers $\lambda$ are chosen from $\{0.0017, 0.0025, 0.0035, 0.0067, 0.0130, 0.0250, 0.050\}$ for MSE and $\{3, 5, 8, 16, 32, 64\}$ for MS-SSIM. All experiments are carried out on NVIDIA A100 GPUs. We set channel dimensions $\{C_1, C_2, C_3, C_4\} = \{128, 192, 256, 320\}$ and set the number of non-linear transform blocks $\{L_1, L_2, L_3\} = \{3, 2, 2\}$. The cluster count is fixed at $64$. The window size of the attention modules is set as 8. The iterative K-Means clustering is updated 5 times per training step, which accounts for only $5\%$ of each step's training time, introducing minimal overhead.

## 4.2 EVALUATION

We test our models on three image datasets: the Kodak dataset (Kodak, 1993) at $768 \times 512$, the Tecnick test set (Asuni et al., 2014) at $1200 \times 1200$, and the CLIC professional validation set (Toderici et al., 2020) at 2K resolution. To assess rate-distortion performance, we report peak signal-to-noise ratio (PSNR), multiscale structural similarity (MS-SSIM), and bits per pixel (bpp). Average bitrate savings are quantified via BD-rate (Bjontegaard, 2001), and average PSNR improvement at matched bitrates via BD-PSNR. Model complexity is evaluated by total parameter count, floating-point operations (FLOPs), peak GPU memory and decoding latency. We also provide detailed encoding latency on A100 and RTX3090 GPUs in Appendix A.14.

## 4.3 RATE-DISTORTION PERFORMANCE

Our CMIC model is benchmarked against the conventional codec VTM-21.0 (Bross et al., 2021), and several SOTA learned methods including ELIC (He et al., 2022), TCM (Liu et al., 2023), MLIC++ (Jiang & Wang, 2023), WeConvene (Fu et al., 2024b), FTIC (Li et al., 2024a), CCA (Han et al., 2024), S2CFormer (Chen et al., 2025a), HPCM (Li et al., 2025c), MLICv2 (Jiang et al., 2025), DCAE (Lu et al., 2025), LALIC (Feng et al., 2025), MambaVC (Qin et al., 2024) and MambaIC (Zeng et al., 2025). The RD curves are shown in Fig. 4-6. Comprehensive BD-rate comparisons, using VTM-21.0 as the reference, are detailed in Tab. 1. Our CMIC model achieves superior performance, reducing BD-rate by 15.91%, 21.34%, and 17.58% on the Kodak, Tecnick, and CLIC datasets.

The proposed CMIC model consistently outperforms leading methods across all evaluated datasets. When measured by BD-PSNR, CMIC surpasses the SOTA Transformer-based FTIC with gains of 0.15 dB (Kodak), 0.32 dB (CLIC), and 0.36 dB (Tecnick). Against the hybrid CNN-Transformer model TCM-L, the superiority is even more pronounced, yielding improvements of 0.28 dB, 0.40 dB, and 0.53 dB on the respective datasets. The model's exceptional performance on high-resolution images (CLIC and Tecnick) validates the powerful global modeling of our CAM module. Furthermore, CMIC also delivers significant MS-SSIM improvements (Fig. 6). It outperforms TCM-L and FTIC by -7.34% and -3.87% respectively, which underscores the overall robustness and superiority of our approach for different metrics. Visual comparisons are provided in Appendix A.5.

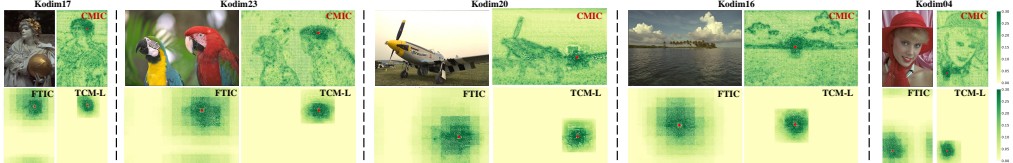

Figure 7: Effective Receptive Field (ERF) visualization for analysis networks in LIC models. Broader and darker green regions denote larger ERFs, which represents a more global spatial context.

Figure 8: Per-image effective receptive field (ERF) visualizations.

Compared with other Mamba-based LIC models, CMIC also demonstrates significant performance gains. It surpasses MambaVC with BD-rate savings of 7.51% on Kodak, 6.80% on CLIC, and 10.09% on Tecnick. Similarly, it outperforms MambaIC by 2.36%, 2.17%, and 6.48% on the three datasets. These consistent advantages across all bitrates on the Tecnick dataset are illustrated in Fig. 1 (c). The rate savings diagrams on the other two datasets are provided in the Appendix A.4. A direct MS-SSIM comparison is omitted because these two competing methods are only optimized for MSE.

## 4.4 MODEL COMPLEXITY

Our proposed CMIC model achieves substantial BD-rate improvements while maintaining moderate complexity levels. Compared to TCM-L (Liu et al., 2023) with a relatively larger parameter scale (75.89M), CMIC is substantially more efficient, reducing FLOPs by 36%, latency by 25%, and peak memory by 43%. In comparison to the SOTA Mamba-based model, MambaIC (Zeng et al., 2025), CMIC also demonstrates clear advantages. Specifically, it reduces parameter count by 56%, FLOPs by 57%, decoding latency by 39%. Notably, it achieves a 78% reduction in GPU memory usage, a significant improvement attributed to its efficient single selective scan rather than the quadratic 2D scans. This favorable complexity-performance balance is primarily attributed to our efficient Mamba architecture, which efficiently captures global context without incurring excessive computational cost. These characteristics underscore CMIC's practical advantages, making it potentially suitable for applications requiring high compression quality coupled with efficient processing.

## 4.5 ABLATION STUDY AND ANALYSIS

We first perform ablation studies to isolate the individual contributions of Content-Adaptive Token Permutation (CTP) and Global-Prior Prompting (GPP), detailed in Tab. 2. Our baseline block, with both components disabled, is equivalent to a vanilla single-scan Mamba block. Our ablation studies focus primarily on the CAM

| CTP | GPP | Kodak | Tecnick | CLIC |
|:---:|:---:|---|---|---|
| | | −13.26 | −17.74 | −14.87 |
| ✓ | | −15.21 | −20.17 | −16.67 |
| | ✓ | −14.27 | −19.13 | −15.34 |
| ✓ | ✓ | **-15.91** | **-21.34** | **-17.58** |

Table 2: Components ablations.

blocks in the transform networks. For the entropy model, adding CAM yields negligible performance gains while increasing latency, indicating a limitation of CAM in enhancing entropy modeling. Detailed results are provided in Appendix A.3.2.

**Content-Adaptive Token Permutation.** CTP uses codebook-based clustering and sequence reordering to group correlated tokens together. Integrating CTP alone yields significant BD-rate reductions of 2.0%, 2.4%, and 1.8% on the Kodak, Tecnick, and CLIC datasets. Conversely, removing CTP from our final model causes a performance drop of 1.6% to 2.2%. This highlights CTP's critical role in overcoming the rigid raster scan, allowing the model

| Method | Throughput ↑ |
|---|---|
| TCM-L | 17.80 |
| MambaVC | 6.55 |
| MambaIC | 9.35 |
| CMIC w/o CTP&GPP | 23.19 |
| CMIC (Ours) | 22.05 |

Table 3: Throughput Ablation.

to enhance interactions between semantically related tokens and capture long-range redundancy.

**Global-Prior Prompting.** Global-Prior Prompting (GPP) mitigates the strict causality of SSM by injecting sample-specific prompts. As shown in Tab. 2, removing GPP from our model reduces the BD-rate gain by 0.7%-1.2%, while adding it to the baseline SSM yields a 0.5%-1.4% improvement. Together, GPP and CTP achieve a total BD-rate reduction of 2.7%-3.6%, demonstrating that the two components are highly complementary.

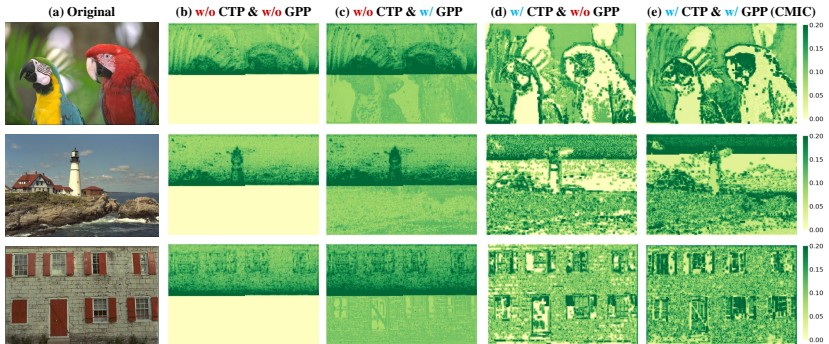

Figure 9: ERF of a single Mamba layer. It shows how GPP introduces non-causality beyond raster scan, and how CTP reshapes ERF toward semantic, content-correlated regions.

**Throughput and Inference Overhead.** Our proposed CTP and GPP modules introduce negligible overhead for both training and inference. As shown in Tab. 3, training throughput on 256×256 patches (batch size 8) slightly decreases from 23.19 to 22.05 samples/s, remaining faster than other Mamba-based models. For inference on a 2K image, decoding time increases by only 4% (from 0.387s to 0.405s), confirming our model's high efficiency.

**Comparison with Other Structures.** To validate the efficacy of our CAM blocks, we substitute them with Conv block and 2D Mamba blocks in stages 3-5 of the model. The results, presented in Tab. 4, demonstrate that CAM blocks yield superior performance over these alternatives while maintaining a comparable parameter count. Furthermore, we compare against variants built purely

|  | BD-rate(%) | Params(M) | FLOPs(T) |
|---|---|---|---|
| Conv Block | −12.89 | 66.47 | 2.18 |
| 2D Mamba | −14.13 | 71.44 | 2.54 |
| Attention-only | −13.06 | 68.23 | 2.33 |
| CAM-only | −14.68 | 70.00 | 2.45 |
| CAM (Ours) | **-15.91** | 69.11 | 2.39 |

Table 4: Ablations on network structures.

with window attention or CAM blocks. Both configurations result in inferior performance, underscoring the effectiveness of our proposed CMIC, in which CAM blocks are strategically integrated.

**ERF Visualization.** We visualize the Effective Receptive Field (ERF) (Luo et al., 2016) of the analysis non-linear transforms of different LIC models in Fig. 7. We calculate the ERF as the magnitude of the gradient of the central latent pixel with respect to the input image, clipping values to [0, 0.20], and averaging over 24 Kodak images. Compared with prior CNN-, Transformer-, and Mamba-based LIC models, CMIC shows a significantly larger receptive field across the entire image.

**Visualization of Content Adaptivity.** We visualize per-image ERFs at fixed query locations (red dots) to further demonstrate the adaptivity of our model. As shown in Fig. 8, CMIC presents global, content-adaptive ERFs whose high-influence regions align with semantic structures and visual intuition about redundancy (e.g., hair/feathers in Kodim17/23, shoreline in Kodim16, aircraft in Kodim20). This reveals that local and distant redundancy is effectively captured and eliminated. In contrast, FTIC and TCM-L yield content-agnostic compact, nearly isotropic ERFs that look similar across images, indicating limited ability to modulate spatial dependencies according to content. More results and comparisons are provided in Appendix A.6.

**Visualization of Non-Causality.** We compute the ERF between the input and output features of a single state-space model layer with soft clustering to analyze the proposed non-causal mechanism and to isolate the effects of our two main components CTP and GPP. As shown in Fig. 9, when both CTP and GPP are removed (column (b)), the tokens are processed in a standard raster-scan order. Taking the center token as the anchor, we observe that the ERF stops exactly at the central position (H//2,W//2): even the subsequent tokens in the same row exhibit zero ERF values. This reflects strict raster-scan causality, where each token is conditioned only on the previously scanned tokens.

In column (c), when we enable GPP, we observe non-zero activations even after the scanned sequence. This shows that GPP allows the state-space model to "see beyond" the strictly causal scan and gain stronger global semantic awareness. Moreover, the activated regions are semantically aligned and meaningful, indicating that the prompt encodes effective global semantic conditions that guide the Mamba scan. In columns (d) and (e), once CTP is applied, the ERF maps no longer exhibit the characteristic raster-scan pattern. Instead, activation spreads over semantically related locations. This demonstrates that CTP effectively breaks the fixed Euclidean-neighbor constraint imposed

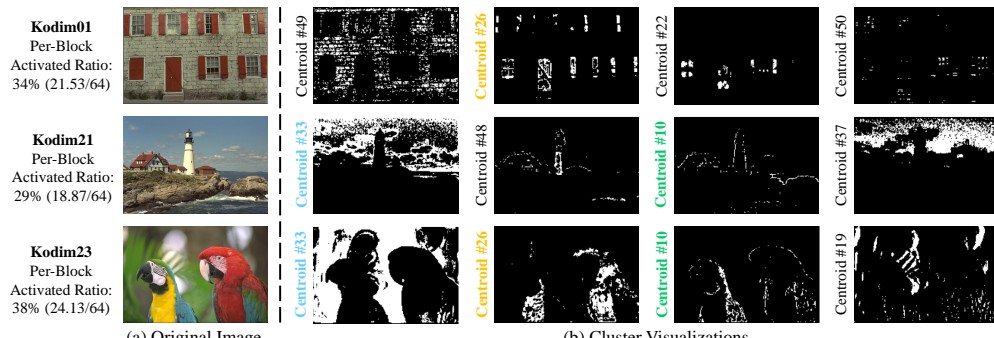

(a) Original Image                                    (b) Cluster Visualizations

Figure 10: Visualization of clustering results from Stage 2 of the analysis network. Each panel displays a binary mask for a single cluster, where white pixels indicate cluster membership.

| Dataset | Mean Activation Count | Activation Ratio | Variance |
|---------|----------------------|------------------|----------|
| Kodak   | 23.27                | 36.37%           | 90.91    |
| CLIC    | 26.30                | 41.10%           | 121.15   |

Table 5: Activation of clusters

| K values | 32 | 64 | 128 |
|----------|----|----|-----|
| BD-rate  | −14.97% | −15.91% | −15.96% |

Table 6: Ablations on K values.

by pre-defined scan orders, prioritizing feature-space proximity over spatial adjacency and further enhancing redundancy elimination between distant yet content-correlated tokens.

**The Adaptivity of K.** We view the centroids of each block as a codebook. While the codebook size remains fixed, the number of active centroids varies adaptively with the content. The codebook size (64) works as an upper bound on the cluster centroids. Although each codebook contains 64 centroids, not all of them are activated during assignment; in practice, many remain empty. Most images typically activate only 16-32 centroids, and the centroids that are active vary from image to image. As shown in Tab. 5, the per-image mean number of activated centroids is 23.27 with a variance of 90.91 on the Kodak dataset. On the CLIC dataset, the corresponding mean is 26.30 with a variance of 121.15. These observations indicate that the exact number of activated clusters depends on image content, so K at inference is effectively dynamic and adaptive to image contents.

**Ablations on Cluster Number.** We also explore the impact of the cluster number $K$, with BD-rates reported for the Kodak dataset (Tab.6). We identify $K = 64$ as a suitable choice. A larger $K$ does not yield much improvement, likely because 64 per-block clusters are sufficient to represent the diversity of most natural images, which also matches our observations about the activated cluster count.

**Cluster Visualization.** We further visualize the clustering results in Fig. 10 to verify the effectiveness of the token permutation mechanism. Each binary mask corresponds to a distinct cluster, and when compared with the original image, it becomes clear that tokens with similar visual or semantic attributes are grouped together (e.g., the red doors and windows in Kodim01, the clouds and sky in Kodim21, and the feathers in Kodim23). This demonstrates that content-similar and semantic-related tokens are reorganized into contiguous sequences, enabling the SSM to efficiently scan neighboring elements in the feature space rather than depending solely on Euclidean distance. Additionally, although the three sample images exhibit different activation rates, they share a subset of activated centroids. We observe that some centroids learn semantically consistent representations: Centroid #10 is mostly triggered by high-gradient edges; #26 tends to respond to red/yellow regions with rich textures; and #33 appears predominantly in smooth blue/green background areas. This suggests that each centroid summarizes a class of high-dimensional visual patterns that are shared across images.

## 5 CONCLUSION

In this work, we introduced Content-Aware Mamba for Image Compression (CMIC), a novel approach tailored to align Mamba-style State-Space models with the inherent two-dimensional structure of image compression. By employing Content-adaptive Token Permutation and Global-Prior Prompting, we enhance SSM's content-awareness to capture long-range redundancy. Specifically, our content-adaptive scanning strategy prioritizes feature-space proximity rather than spatial arrangement. Embedding global priors via the prompt dictionary relaxes the strict causality without quadrupling the computational load. Consequently, CMIC efficiently eliminates distant redundancy while retaining linear computational complexity. Experimental results demonstrate that CMIC achieves SOTA RD performance and consistently outperforms recent Mamba-based image compression models.

ACKNOWLEDGMENT

This work was supported by the National Key Research and Development Program of China under Grant 2024YFF0509700, National Natural Science Foundation of China(62471290,62431015,62331014) and the Fundamental Research Funds for the Central Universities.

ETHICS STATEMENT

All the authors read and adhere to the ICLR Code of Ethics.

REPRODUCIBILITY STATEMENT

We ensure the experiments in this paper are reproducible. We provide detailed settings about datasets and training in Section 4.1.

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

# A APPENDIX

## A.1 VTM SETTINGS

We provide the corresponding bash commands for encoding and decoding an image using VTM-21.0 (Browne et al.). The resulting YUV file can then be further processed or converted into RGB format for distortion calculation.

**Encoding**

```
VTM−21.0/ bin / EncoderAppStatic −i tmp.yuv −c VTM−21.0/ cfg /
    encoder_intra_vtm.cfg −q 61 −o /dev/null −b tmp.bin −wdt 768 −
    hgt 512 −fr 1 −f 1 −−InputChromaFormat=444 −−InputBitDepth=8
    −−ConformanceWindowMode=1
```

**Decoding**

```
VTM−21.0/ bin / DecoderAppStatic −b tmp.bin −o tmp.yuv −d 8
```

## A.2 COMPARISON WITH ANOTHER CLUSTERING-BASED LIC MODEL

In comparison with another CNN-based clustering-based LIC model Zhang et al. (2024b), we emphasize that the key distinctions lie primarily in our motivation to overcome the raster-scan and causality limitations inherent in the Mamba-based LIC model, as well as our use of a permutation-equivariant, fine-grained clustering strategy.

### A.2.1 OUR MOTIVATIONS AND INSIGHTS

**First Insight.** Vanilla Mamba's fixed raster-scan order limits effective redundancy removal, as optimal compression requires interactions among semantically related tokens. Although Mamba has global modeling capabilities, its raster-scan approach places unrelated tokens adjacently, which limits efficient interactions between correlated tokens. To address this, we propose Content-Adaptive Token Permutation (CTP), which clusters and rearranges tokens so that Mamba prioritizes interactions among similar tokens. Specifically, CTP:

- Clusters tokens into correlated groups (validated by Fig. 10).
- Reorders tokens to position related ones adjacently, improving redundancy elimination.

Our CAM block thus enhances token interactions and state-space updates, significantly boosting compression efficiency.

**Second Insight.** Vanilla Mamba's strict causality, where tokens depend only on previously scanned tokens, limits its vision task performance. To address this, we propose a sample-specific clustering-based prompt to introduce global context and relax causality constraints efficiently. Unlike prior methods that rely on costly multi-directional scanning, we:

- Generate a sample-specific prompt from clustering results and a distribution-aware prompt dictionary.
- Use this prompt to incorporate global statistics into the state-space (e.g., matrix $C$), capturing sample-specific prior.

Thus, our method provides global context and mitigates Mamba's causality constraints without extra multi-directional scanning.

### A.2.2 CLUSTERING COMPARISON

The clustering strategy in Zhang et al. (2024b) still partially relies on spatial relationships and is a coarse-level clustering. In contrast, our approach introduces a fundamentally different clustering methodology for fine-grained, non-Euclidean grouping. We can summarize the two methods in short as follows:

- **Zhang et al. (2024b)'s Method:** Uses average pooling to generate $2 \times 2$ cluster centers, relying on spatial relationships and a checkerboard mask for even-numbered blocks.
- **Our Method:** Uses a codebook for center lookup, determining clusters based on token content without relying on spatial coordinates.

The two methods demonstrate two main distinct characteristics:

**Spatial Anchoring vs. Non-Euclidean Adaptive Clustering** Zhang et al. (2024b) is grid-anchored ($2 \times 2$ average-pooled centers + checkerboard), so assignments change under pixel permutations or shifts. Our clustering is permutation-equivariant and translation-invariant; assignments are unaffected by rearrangements.

**Coarse Pooling vs. Fine-grained Clustering** The average-pooled centers in Zhang et al. (2024b) are biased toward homogeneous regions (e.g., background), reducing discriminative power and yielding coarse groups. We use a codebook of semantic prototypes (similar to codebooks in VQ-VAE [2]) queried by CAM blocks, enabling precise, fine-grained grouping needed for compression. As shown in Figure 8, our clustering yields fine-grained groupings, capturing details such as parrot feathers and cloud structures.

## c) PERFORMANCE COMPARISON

According to the reported RD curves, Zhang et al. (2024b) achieves BD-rate improvements of $-8.75\%$ and $-9.64\%$ over VTM-21.0 on Kodak and Tecnick, respectively, which is lower than our BD-rate gains of -15.91% (Kodak) and -21.34% (Tecnick).

In summary, we acknowledge the contribution of Zhang et al. (2024b) to the clustering-based LIC model. Our work, however, has a totally different motivation and employs a completely different design of clustering.

## A.3 MORE DETAILS AND ABLATIONS ABOUT ENTROPY MODEL

### A.3.1 DETAILS OF ENTROPY MODEL

The Space-Channel ConTeXt (SCCTX) model (He et al., 2022) is a fast and parallel entropy model for learned image compression. Given latent features $\hat{\mathbf{y}} \in \mathbb{R}^{M \times H \times W}$, SCCTX splits the $M$ channels into $K$ groups, indexed by $k$. Early groups have fewer channels to focus on high-energy components, while later groups have more channels for low-energy components. For each group $k$ and each position $i$, SCCTX predicts the entropy parameters $\Theta_i^{(k)} = (\mu_i^{(k)}, \sigma_i^{(k)})$ using three types of context as follows in Fig. A.3.

First of all, the spatial context $\Phi_{\text{sp},i}^{(k)}$ is computed by a network $g_{\text{sp}}^{(k)}$ from previously decoded symbols in the same group:

$$\Phi_{\text{sp},i}^{(k)} = g_{\text{sp}}^{(k)} \left( \hat{\mathbf{y}}_{<i}^{(k)} \right).$$

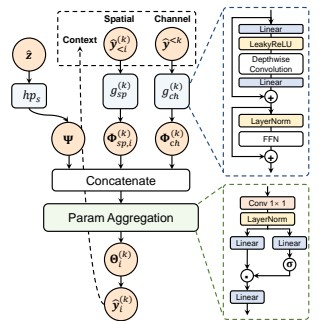

The channel context $\Phi_{\text{ch}}^{(k)}$ is computed by another network $g_{\text{ch}}^{(k)}$ from all symbols in previous groups:

$$\Phi_{\text{ch}}^{(k)} = g_{\text{ch}}^{(k)} \left( \hat{\mathbf{y}}^{(<k)} \right).$$

The hyperprior context $\Psi$ is obtained by applying a nonlinear transform function $hp_s$ to the hyperprior latent $\hat{\mathbf{z}}$:

$$\Psi = hp_s(\hat{\mathbf{z}}).$$

All three contexts are combined and passed through a simple aggregation network, which outputs the final entropy parameters:

$$\Theta_i^{(k)} = \text{Aggregation} \left( \Phi_{\text{sp},i}^{(k)}, \Phi_{\text{ch}}^{(k)}, \Psi \right).$$

By combining these contexts in parallel, SCCTX can use both local and global information, which leads to fast decoding and good compression results.

Figure 11: Overview of Our Enhanced Efficient SCCTX Entropy Model.

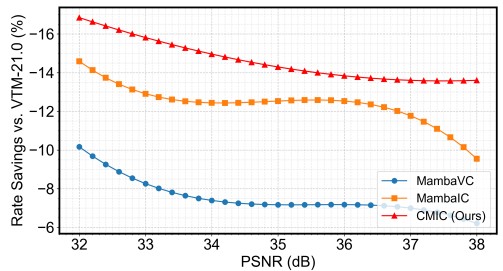
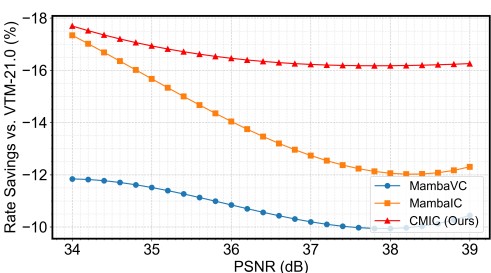

Figure 12: Rate Savings on the Kodak dataset.   Figure 13: Rate Savings on the CLIC dataset.

### A.3.2 ABLATIONS ON ENTROPY MODEL

We provide three more ablation studies about the entropy model to demonstrate the superiority of our CAM block.

First, we compare our enhanced Efficient SCCTX model with the Conv SCCTX baseline (removing both FFN in $g_{\mathrm{ch}}^{(k)}$ and gated mechanism in Param Aggregation). The results are detailed in Tab. 7. We emphasize that our entropy model is not intended to be state-of-the-art. It is a lightweight variant built on SCCTX with a few additional MLP layers, aiming for a favorable speed-performance trade-off. Despite its simplicity, the model demonstrates a slight improvement over the Conv SCCTX.

Second, we also tried to apply our CAM block to replace the combination of Depth-wise convolution and FFNs in the $g_{\mathrm{ch}}^{(k)}$. This variant is denoted as "CMIC (CAM SCCTX)" in Tab. 7. Although this variant offers some advantages on high-resolution datasets, the improvements are modest, and it increases decoding latency from 0.405 s to 0.471 s, a slowdown about 16%. Accordingly, we use the efficient, enhanced SCCTX variant in our CMIC model, while clustering mechanisms and the Mamba architecture remain promising directions for future work on entropy modeling.

These results demonstrate that, while content-adaptive processing and Mamba's global modeling capabilities are crucial for the non-linear transform networks, this strong global awareness does not offer significant benefits for the entropy model, which is used to model the distribution of the latents after redundancy removal. In contrast, depth-wise convolution, which emphasizes local relationships, is a highly efficient and sufficiently powerful operator. Additionally, the extra linear and MLP layers we inserted are also proven to be both efficient and powerful for a multi-layer slicing entropy model.

Table 7: Comparison of different SCCTX-based Entropy models.

| Model | BD-rate [Kodak] | BD-rate [CLIC] | BD-rate [Tecnick] |
|---|---|---|---|
| CMIC (Our Enhanced SCCTX) | -15.91% | -17.58% | -21.34% |
| CMIC (Conv SCCTX) | -15.11% | -16.78% | -20.53% |
| CMIC (CAM SCCTX) | -15.81% | -17.73% | -21.45% |

Third, we replace the entropy model in our CMIC with the advanced T-CA entropy model used in FTIC. Despite using the same entropy model, our approach still outperforms the state-of-the-art FTIC, demonstrating the effectiveness of our transform networks.

Table 8: Performance of CMIC when using the advanced T-CA entropy model from FTIC.

| Model | BD-rate [Kodak] | BD-rate [CLIC] | BD-rate [Tecnick] |
|---|---|---|---|
| FTIC | -12.94% | -10.21% | -13.89% |
| CMIC (with FTIC entropy model) | -15.67% | -17.56% | -21.42% |

## A.4 More Rate Saving Comparisons

We also provide plots of Rate Savings vs. PSNR on the Kodak and CLIC datasets in Fig. 12 and Fig. 13. Those two figures and Fig. 1 (c) show that our CMIC model steadily outperforms the MambaVC Qin et al. (2024) and MambaIC Zeng et al. (2025) models across all three datasets with different resolutions, at different bitrates. This demonstrate the superiority of our CMIC model.

## A.5 Visual Comparisons

We provide more visual results in Fig. 14-15, comparing with traditional codec VTM (Browne et al.) and advanced LIC model TCM (Liu et al., 2023).

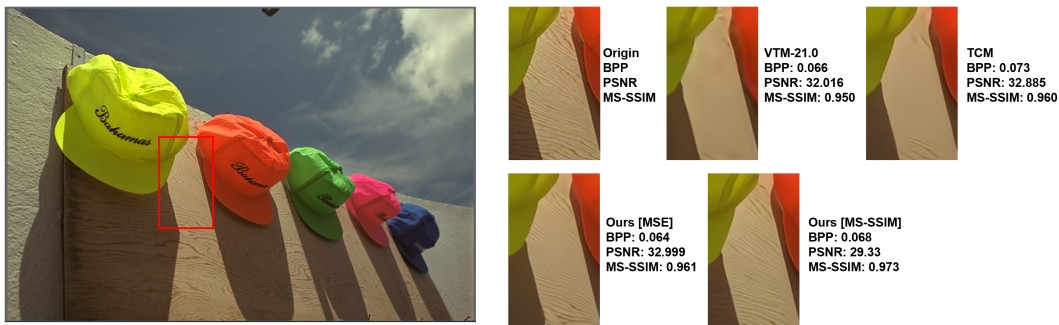

Figure 14: Visual Comparisons on Kodim03.

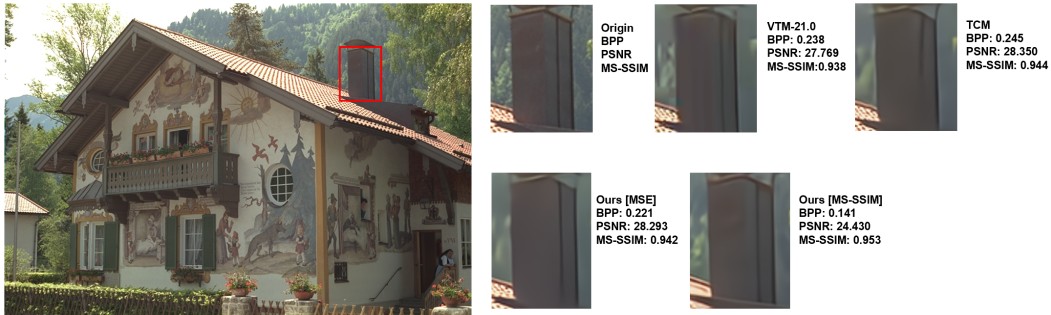

Figure 15: Visual Comparisons on Kodim24.

## A.6 More Visualizations of Per-Image ERFs

We provide more visualizations of per-image erfs in Fig. 16. Our model demonstrates much more content-adaptive ERFs than CNN-based, Transformer-based and even Mamba-based LIC models. While MambaIC demonstrates a degree of global awareness, its Effective Receptive Field (ERF) exhibits distinct artifacts. Specifically, it manifests as cross-shaped regions of high gradients centered on the target pixel (the red dot). This artifact is content-agnostic and exist in every image. It may result from its four-directional scanning mechanism. This causes the model to focus on capturing irrelevant information. In contrast, our model concentrates on capturing genuinely redundant information within images, which is more adaptive and effective.

## A.7 More Basic Block Structures

We detail the architectures of the Attention Block and Conv Block in Fig. 17, both designed to efficiently capture local contextual information.

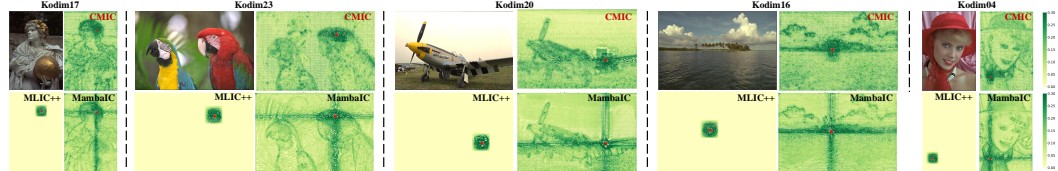

Figure 16: More Per-image effective receptive field (ERF) visualizations.

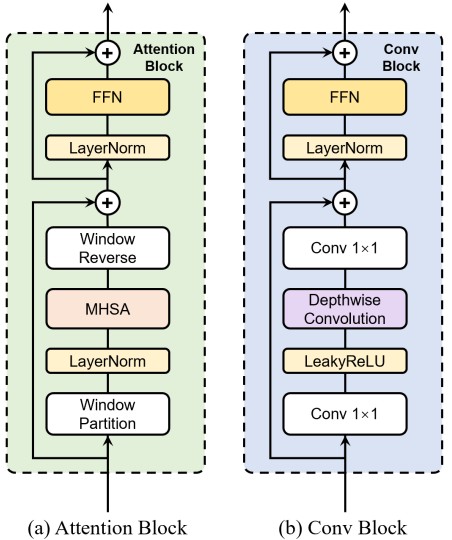

(a) Attention Block     (b) Conv Block

Figure 17: Details of Attention and Conv block

**Attention Block.** For an input feature map $\mathbf{x}$, we first split it into small windows. Each window is normalized (LayerNorm, LN) and processed by Multi-Head Self-Attention (MHSA). The outputs are then combined back together and added to the original input (residual connection). Finally, we apply another normalization, a feed-forward network (FFN), and another residual addition.

**Conv Block.** Replacing the MHSA stem, we use an efficient depthwise-separable convolution stack: $1\times1$ point-wise conv $\rightarrow$ depthwise conv $\rightarrow$ LeakyReLU $\rightarrow$ $1\times1$ point-wise conv. The remaining LN–FFN sequence and the two residual shortcuts are identical to the Attention Block, enabling a lightweight yet expressive local-context modeling.

## A.8 TRAINING STABILITY

Since K-means clustering and token sorting are non-differentiable, the resulting gradients are biased. However, our EMA and codebook-based K-means updates ensure stable convergence when training the CMIC model.

To demonstrate stable training and convergence, we plot test-loss curves versus training steps for our model under different random seeds and for the ELIC baseline He et al. (2022). As shown in Fig. 18, our model is insensitive to the choice of random seed and centroid initialization (the centroids are initialized simply as the average of token embeddings within each segment from the first batch), and it converges reliably.

## A.9 RATIONALE FOR PER-BLOCK CLUSTER CODEBOOK

Each CAM block maintains its own K cluster centers and does not share the centers.

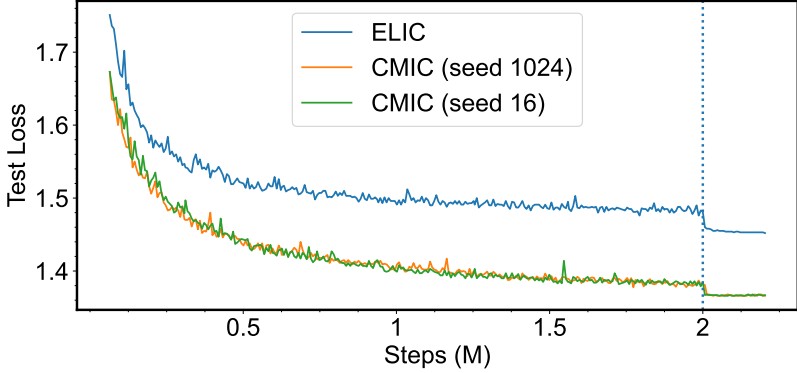

Figure 18: Test losses vs. Training Steps.

Sharing cluster centers across all blocks can lead to significant alignment challenges, as feature distributions may vary drastically. Specifically, features at different stages often exhibit entirely distinct resolutions, receptive fields, and semantic levels. Additionally, channel widths may differ between stages, necessitating extra projection layers for dimensional alignment—this inevitably introduces additional parameters. A single shared set of cluster centers would be forced to accommodate heterogeneous feature spaces, potentially compromising performance.

By allowing each block to learn its own dedicated codebook of cluster centers, the centers can specialize according to local feature statistics, resulting in more stable training and higher-quality representations.

**Parameter impact.** The total parameter count for all eight codebooks of centers is 114688, which accounts for 0.166% of the total parameter count (69.11M). This marginal overhead suggests the codebooks introduce minimal additional parameters.

## A.10   THE ORDERING OF TOKENS WITHIN A CLUSTER AND THE ORDERING OF CLUSTERS

We utilize stable sorting and:

1. The tokens within a single cluster are processed in their original raster-scan order.

2. The cluster order is determined by the order of centers in the center codebook.

## A.11   CODE AVAILABILITY

The full implementation code and checkpoints will be released immediately upon acceptance.

## A.12   THE USE OF LARGE LANGUAGE MODELS (LLMs)

We utilized LLMs only for minor linguistic refinement. All the main contributions, including research ideas, experimental design, and data analysis, are the original work of the authors.

### A.13 DETAILED DISCUSSIONS ABOUT MAMBAIRV2

In this section, we provide detailed comparisons and visualization analysis about our CAM block and MambaIRv2 (Guo et al., 2024a).

We emphasize that our method differs from MambaIRv2 at a fundamental level in both *motivation* and *technical design*. Accordingly, we provide detailed comparisons and visual evidence to demonstrate why our approach is better suited to learned image compression.

#### A.13.1 MOTIVATION COMPARISON.

Our goal is to enhance Mamba's ability to eliminate redundancy among tokens that are content-correlated yet spatially distant. To this end, our CAM proposes **redundancy-aware clustering** and **redundancy-aware prompts**, enabling the model to identify global redundancy and eliminate it more effectively.

In contrast, MambaIRv2 is developed for image super-resolution; its architecture is not designed to discover or model redundancy for compression.

**This task mismatch naturally leads to different classification and prompting mechanisms.**

#### A.13.2 TECHNICAL COMPARISONS.

We make technical comparisons in two aspects as follows.

1. **Token classification.**
   - **Our CAM: redundancy-aware clustering.**
     We classify tokens via codebook-based KMeans clustering, which is explicitly tailored for redundancy discovery. Using cosine similarity, we perform iterative mini-batch KMeans over token features to learn a shared codebook of centroids. These centroids group content-similar tokens and enable their reordering, so redundancy among semantically similar but spatially distant regions can be reduced effectively.
   - **MambaIRv2: unsupervised classification head**
     MambaIRv2 employs a lightweight linear projection head to produce per-token logits, followed by Gumbel-Softmax sampling for discrete assignments. This head is trained without explicit semantic or redundancy-oriented supervision, so the resulting clusters lack interpretability and do not reliably reflect token redundancy.

2. **Global prompting.**
   - **Our CAM: redundancy-aware dictionary**
     While our prompting follows the Attentive State Space equation introduced in MambaIRv2, our prompts are *strictly tied to redundancy-aware clustering*. Each prompt vector is projected from a specific cluster centroid and thus inherits clear semantic meaning. As specified in Sec. 3.4, we index the prompt dictionary using the cluster assignments, so the prompt signal explicitly encodes a global redundancy distribution over semantic clusters, highlighting heavily redundant clusters for the Mamba block to emphasize.
   - **MambaIRv2: standalone dictionary**
     MambaIRv2 learns a prompt pool, which is a standalone learnable matrix optimized directly by gradient descent without explicit semantic constraints and without coupling to any redundancy-aware clustering. Consequently, its prompts do not explicitly represent global redundancy across content clusters.

**Summary of the design gap.** Our method is purpose-built for the central requirement of learned image compression: **capturing and eliminating redundancy**. Therefore, we introduce (i) a redundancy-aware clustering strategy specialized for redundancy detection, and (ii) a cluster-tied prompt conditioning mechanism that provides the state-space model with a global redundancy-distribution signal. MambaIRv2, designed for super-resolution, does not incorporate these redundancy-oriented components.

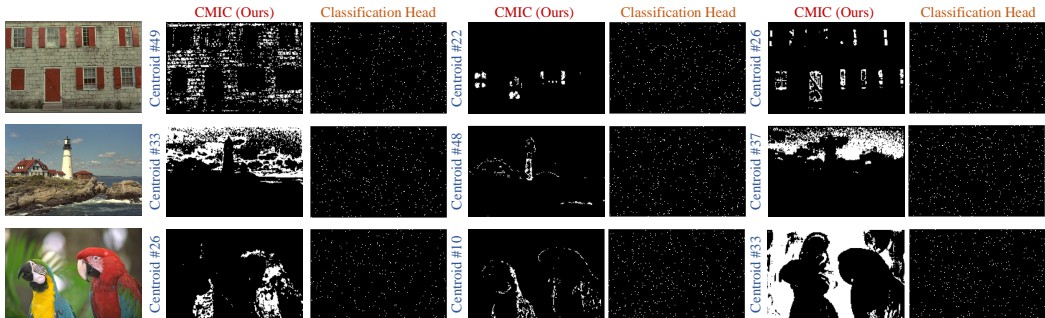

Figure 19: Visualizations of classification results.

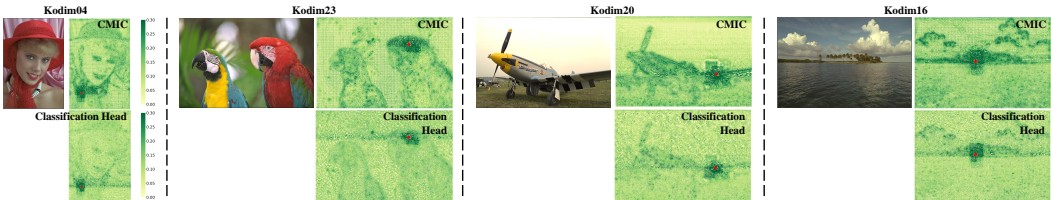

Figure 20: Per-image effective receptive field comparisons.

### A.13.3 EXPERIMENTAL EVIDENCE.

We conduct ablations on the classification and dictionary strategies of MambaIRv2 within our framework. Both qualitative and quantitative comparisons are provided to highlight the differences between our redundancy-aware token classification and redundancy-aware dictionary and those of MambaIRv2.

1. **Classification Results.** We first compare the classification strategies: our codebook-based KMeans clustering versus the unsupervised linear classification head in MambaIRv2.

   - As shown in Fig. 19, our redundancy-aware clusters align well with semantic structures and visual redundancy patterns, whereas the assignments produced by MambaIRv2's classification head do not correspond to semantic content or redundancy.
   - Moreover, Fig. 21 visualizes all 64 clusters produced by the classification-head strategy; none exhibits meaningful redundancy or semantic consistency.

   These observations indicate that MambaIRv2's routing is not designed for compression, while our clustering is explicitly tailored to capture redundancy.

2. **Single-image ERF Visualization.** Fig. 20 in the revision further provides single-image ERF visualizations.

   - Our ERF concentrates sharply on redundant, content-correlated tokens and regions, including spatially distant areas, demonstrating strong content adaptivity.
   - In contrast, MambaIRv2 exhibits a more content-agnostic long-range pattern: responses are relatively uniformly spread over the entire image, with much weaker sensitivity to semantic structure and redundancy.

   This shows that our CAM captures redundancy more effectively, which in turn leads to improved compression performance.

3. **BD-rate Comparison.** We implement two variants to quantitatively compare the classification and prompt-pool strategies within our compression framework. The results are in Tab. 9.

   - The first variant, *CMIC (w/ Standalone Dictionary)*, replaces our redundancy-aware dictionary with the standalone dictionary in MambaIRv2.

- The second variant, *CMIC (w/ Classification Head)*, replaces our redundancy-aware clustering with the unsupervised classification head from MambaIRv2. Since our redundancy-aware dictionary relies on the clustering centers, for *CMIC (w/ Classification Head)* we use the standalone dictionary for prompt conditioning.

The resulting BD-rate comparisons show that our CAM design consistently outperforms the MambaIRv2-style variants, confirming the benefit of our redundancy-aware classification and dictionary.

| Model | Kodak | Tecnick | CLIC |
|---|---|---|---|
| CMIC (CAM) | -15.91 | -21.34 | -17.58 |
| CMIC (w/ Standalone Dictionary) | -15.02 | -20.21 | -16.73 |
| CMIC (w/ Classification Head) | -14.46 | -19.76 | -15.83 |

Table 9: BD-rate comparison between our CAM block and the MambaIRv2 designs.

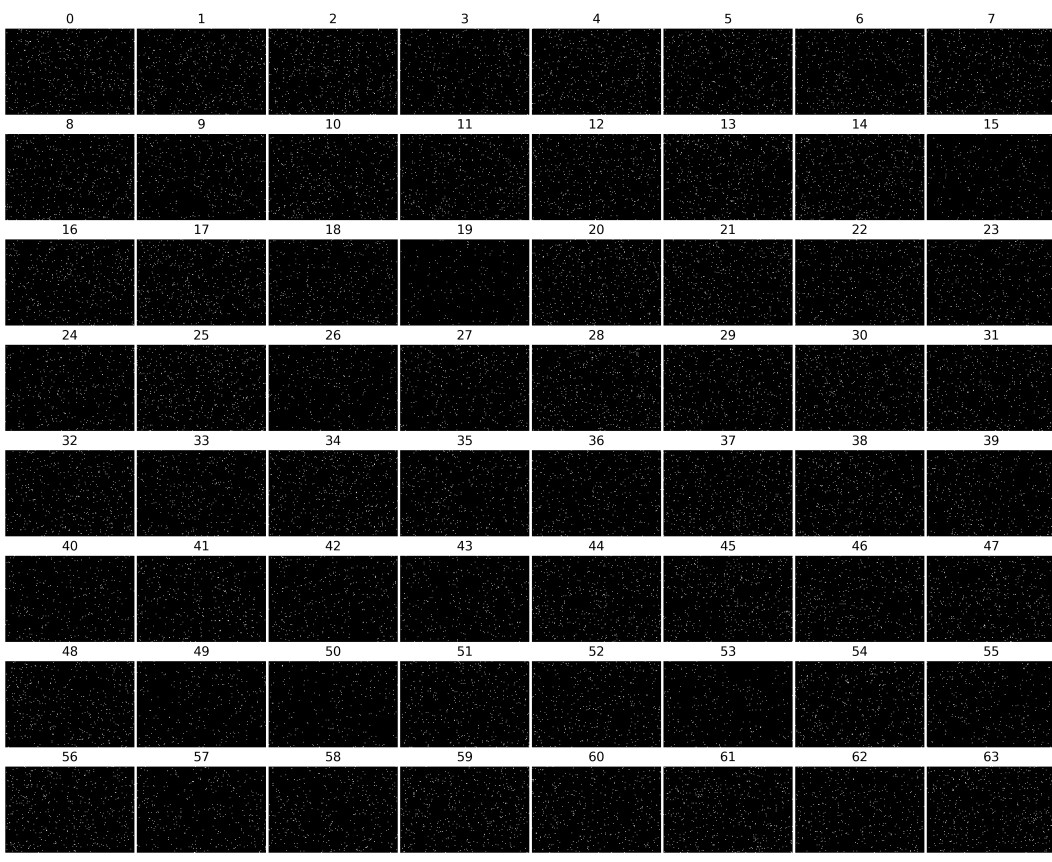

Figure 21: All 64 Cluster Visualizations of Linear Classification Head Strategy.

| Model | A100-Enc-Lat. (s) | A100-Dec-Lat. (s) | 3090-Enc-Lat. (s) | 3090-Dec-Lat. (s) |
|---|---|---|---|---|
| ELIC | 0.583 | 0.335 | 0.671 | 0.392 |
| MLIC++ | 0.508 | 0.547 | 0.601 | 0.611 |
| WeConvene | 1.264 | 1.293 | 1.332 | 1.373 |
| TCM-L | 0.647 | 0.542 | 0.701 | 0.612 |
| CCA | 0.526 | 0.385 | 0.599 | 0.418 |
| FTIC | $> 10$ | $> 10$ | $> 10$ | $> 10$ |
| S2CFormer | 0.755 | 0.362 | 0.891 | 0.443 |
| HPCM | 0.282 | 0.304 | 0.335 | 0.362 |
| LALIC | 0.909 | 0.385 | 0.990 | 0.460 |
| DCAE | 0.469 | 0.478 | 0.551 | 0.543 |
| MambaVC | 1.223 | 0.425 | 1.390 | 0.502 |
| MambaIC | 1.436 | 0.669 | 1.572 | 0.783 |
| CMIC (Ours) | 0.618 | 0.405 | 0.692 | 0.464 |

Table 10: Encoding/decoding latency on A100 and RTX3090 (seconds)

## A.14 ENCODING LATENCY ON DIFFERENT GPUS

We provide encoding / decoding latency of different SOTA models on both A100 and RTX 3090 GPUs. As shown in Tab. 10, our model achieves much lower encoding latency and decoding latency compared to previous Mamba-based LIC models (for both A100 and 3090), demonstrating the effectiveness of our high-efficiency adaptive strategies.

