# OpenReview forum: "Content-Aware Mamba for Learned Image Compression"
_ICLR.cc/2026/Conference — ICLR 2026 Poster_

### Official Review · Reviewer_ysic · 2025-10-27

**Soundness:** 3
**Presentation:** 3
**Contribution:** 3
**Rating:** 6
**Confidence:** 4

**Summary:**

The paper proposes Content-Aware Mamba (CAM) for learned image compression. This paper proposes two contributions: (1) content-adaptive token permutation via a codebook/K-means clustering that reorders tokens to put feature-similar patches adjacent before a 1-D selective scan; and (2) global-prior prompting, which injects sample-specific “prompts” derived from cluster assignments into the Mamba output projection to relax strict causality. On Kodak, Tecnick, and CLIC, CMIC claims strong BD-rate gains over VTM-21.0 and recent LIC models, with moderate compute and memory.

**Strengths:**

1. Clear motivation & design: Shows how fixed raster scans and causality in SSMs are misaligned with image structure; proposes permutation + prompting to address both.
2. Empirical results: Solid BD-rate improvements on three standard datasets; Comparable complexity numbers (params/FLOPs/latency/memory).

**Weaknesses:**

1. Possible shape mismatch: The prompting equation
    \[
    O_i = (C + P) h_i + D x_i
    \]
    leaves the shape of $P$ ambiguous. Earlier $P \in \mathbb{R}^{N \times d_s}$ (per-token prompt vectors), whereas $C \in \mathbb{R}^{d \times d_s}$. Adding them is dimensionally inconsistent unless $P$ is lifted to $\mathbb{R}^{d \times d_s}$ per token or used to gate $C$ via a mapping. Please clarify this precisely (e.g., broadcast).

2. Report encoding latency (not only decoding). Only decoder latency on A100 is provided.

3. The paper does not justify this permutation design choice. It would be strengthened by a comparison to simpler, end-to-end differentiable alternatives for learning a permutation, such as using attention scores for sorting, Gumbel-Softmax sampling, or an optimal transport-based approach.

4. The ablation study in Appendix A.3.2 shows that replacing convolutions with CAM blocks in the entropy model provides negligible gains and increases latency. While the authors conclude that the entropy model benefits more from local context, this finding is interesting and slightly undermines the generality of the CAM block. A brief mention of this limitation in the main paper's ablation study would provide a more balanced perspective.

5. The core idea of clustering tokens for content-adaptive processing is not entirely new. The authors cite Zhang et al. (2024), which also uses a clustering scheme. While the authors provide a good discussion in Appendix A.2, differentiating their fine-grained, non-Euclidean clustering from the coarse, grid-anchored approach of Zhang et al., this critical comparison is relegated to the appendix. For a top-tier conference, this discussion of related work and key differentiators should be integrated into the main paper.

**Questions:**

Please see the weakness part. Overall, I find the paper satisfying. The paper would be further improved if the authors addressed the weaknesses above.

---

> ### Author Response · Authors · 2025-11-22
> **Response to Reviewer ysic**
>
> ### 📝 Q1: "Possible shape mismatch"
>
> **Response:**
>
> The dynamic parameter $c$ produced by the SSM projection has shape
> $$
> C \in \mathbb{R}^{K \times d_s \times N},
> $$
>
> with \(K=1, only 1 scan\) in this work.
>
> We reshape the token-wise prompt $P$ to
> $$
> P \in \mathbb{R}^{1 \times d_s \times N},
> $$
> and add it to $C$ with broadcasting along the scan dimension K, i.e.,
> $$
> \tilde{C} = C + \text{broadcast}(P).
> $$
> Therefore, $C$ and $P$ are dimensionally consistent and are added element-wise per token.
>
> ---
>
> ### 📝 Q2: "Report encoding latency (not only decoding). Only decoder latency on A100 is provided."
>
> **Response:**
>
> Thank you for your advice. We report encoding/decoding latencies on A100 and 3090 GPUs here.
>
> | Model       | A100-Enc-Lat. (s) | A100-Dec-Lat. (s) |      | 3090-Enc-Lat. (s) | 3090-Dec-Lat. (s) |
> | ----------- | ----------------- | ----------------- | ---- | ----------------- | ----------------- |
> | ELIC        | 0.583             | 0.335             |      | 0.671             | 0.392             |
> | MLIC++      | 0.508             | 0.547             |      | 0.601             | 0.611             |
> | WeConvene   | 1.264             | 1.293             |      | 1.332             | 1.373             |
> | TCM-L       | 0.647             | 0.542             |      | 0.701             | 0.612             |
> | CCA         | 0.526             | 0.385             |      | 0.599             | 0.418             |
> | FTIC        | >10               | \>10              |      | >10               | >10               |
> | S2CFormer   | 0.755             | 0.362             |      | 0.891             | 0.443             |
> | HPCM        | 0.282             | 0.304             |      | 0.335             | 0.362             |
> | LALIC       | 0.909             | 0.385             |      | 0.990             | 0.460             |
> | DCAE        | 0.469             | 0.478             |      | 0.551             | 0.543             |
> | MambaVC     | 1.223             | 0.425             |      | 1.390             | 0.502             |
> | MambaIC     | 1.436             | 0.669             |      | 1.572             | 0.783             |
> | CMIC (Ours) | 0.618             | 0.405             |      | 0.692             | 0.464             |
>
> Our model achieves much lower encoding latency and decoding latency compared to previous Mamba-based LIC models, demonstrating the effectiveness of our high-efficiency adaptive strategies. The similar phenomenon is observed for both A100 and RTX 3090 GPUs.
>
> We add it to the Table 9 in A.14 ENCODING LATENCY ON DIFFERENT GPUS.

---

> ### Author Response · Authors · 2025-11-22
> **Response to Reviewer ysic**
>
> ### 📝 Q3: "The paper does not justify this permutation design choice."
>
> **Response:**
>
> Thank you for your advice. Since our model does not include a global attention module, we cannot obtain global attention scores for sorting. Instead, we adopt a lightweight learned linear classification head combined with Gumbel-Softmax, which serves as an end-to-end differentiable alternative for clustering.
>
> Here is a BD-rate comparison. We add this ablation to demonstrate the semantically meaningful and effectiveness of our clustering strategy. As shown in the table, the variant with unsupervised linear head can only slightly outperform "CMIC w/o CTP" as it although can help to mitigate the non-causality but can hardly provide redundancy-aware clustering for better capturing and removing redundancy. In other words, it is not well suited to image compression, whereas our codebook-based clustering is specifically designed to better eliminate redundancy in learned image compression.
>
>
>
> | Model                         | Kodak  | Tecnick | CLIC   |
> | :---------------------------- | :----- | :------ | :----- |
> | CMIC                          | -15.91 | -21.34  | -17.58 |
> | CMIC+LinearHead&GumbelSoftmax | -14.46 | -19.76  | -15.83 |
> | CMIC w/o CTP                  | -14.27 | -19.13  | -15.34 |
>
> ---
>
> ### 📝 Q4: "A brief mention of this limitation (CAM for entropy model) in the main paper's ablation study would provide a more balanced perspective."
>
> **Response:**
>
> Thank you for your suggestion. We add this claim to Section 4.5 ABLATION STUDY AND ANALYSIS (Line 416-419).
>
> ---
>
> ### 📝 Q5: "This discussion of related work and key differentiators should be integrated into the main paper."
>
> **Response:**
>
> Thank you for your suggestion. We add a brief discussion to Section 2.3 CONTENT-ADAPTIVE IMAGE COMPRESSION (Line144-148). But due to space limitation, we have to put the detailed discussion in the appendix.

---

> > ### Comment · Reviewer_ysic · 2025-11-26
> >
> > I thank the authors for their detailed responses and the revisions made to the paper. I have reviewed the rebuttal and the updated manuscript, and I find that my concerns have been addressed. I will maintain my positive score.

---

> > > ### Author Response · Authors · 2025-11-26
> > > **Response to Reviewer ysic**
> > >
> > > We sincerely thank the reviewer for the positive evaluation and for the constructive comments that helped improve our paper. We greatly appreciate your time and consideration.

---

### Official Review · Reviewer_QkwG · 2025-10-29

**Soundness:** 3
**Presentation:** 3
**Contribution:** 1
**Rating:** 2
**Confidence:** 4

**Summary:**

This paper addresses the issues of standard Mamba-style state-space models (SSMs) in learned image compression (LIC), namely content-agnostic fixed scanning (failing to effectively associate tokens that are spatially distant but content-similar) and strict causality (solved by multi-directional scanning yet with quadrupled complexity). It proposes Content-Aware Mamba (CAM), which relies on two core innovations: content-adaptive token permutation (reordering tokens by feature-space proximity via a shared learnable codebook) and global-prior prompting (injecting sample-specific global priors to relax causality).

**Strengths:**

1. The overall writing of the paper is clear, and the figures are well-presented.
2. The proposed method achieves the optimal experimental performance.

**Weaknesses:**

1. It is necessary to clarify the specific differences between the proposed method and Mambairv2 [1].
Both content-aware selective scanning and Learnable Prompt have been proposed in Mambairv2.
Though I understand that Mambairv2 and the proposed CMIC are applied to two different tasks, the designs in this paper are mostly borrowed from Mambairv2 in a direct manner. There is insufficient technical contribution or further insights beyond Mambairv2 and specifically benefits image compression.
2. The comparison with the latest approaches, such as MLICv2 [2], is missing.

---
[1] Guo, Hang, et al. "Mambairv2: Attentive state space restoration." Proceedings of the Computer Vision and Pattern Recognition Conference. 2025.

[2] Jiang, Wei, et al. "MLICv2: Enhanced Multi-Reference Entropy Modeling for Learned Image Compression." arXiv preprint arXiv:2504.19119 (2025).

**Questions:**

See the Weaknesses for details.

---

> ### Author Response · Authors · 2025-11-22
> **Response to Reviewer QkwG**
>
> ### 📝 Q1: "The specific differences between the proposed method and MambaIRv2"
>
> **Response:**
>
> Thank you for your question.
>
> We emphasize that our method and MambaIRv2 differ fundamentally in both motivation and technical design. Accordingly, we provide **detailed comparisons and visual evidence in Appendix A.13 DETAILED DISCUSSIONS ABOUT MAMBAIRV2** to demonstrate the superiority of our approach for image compression.
>
> In short:
>
> - Our method is purpose-built for compression: it targets redundancy elimination via redundancy-aware token clustering and a cluster-tied redundancy-aware prompt dictionary.
> - MambaIRv2 is designed for super-resolution and uses an unsupervised classification head and a standalone prompt pool without semantic/redundancy guidance.

---

> ### Author Response · Authors · 2025-11-22
> **Response to Reviewer QkwG**
>
> (Continued)
>
> In detail:
>
> > **Motivation Comparison**
> >
> > Our core motivation is to enhance Mamba’s ability to eliminate redundancy between tokens that are content-correlated but spatially distant. To achieve this, our CAM explicitly couples redundancy-aware clustering with redundancy-aware dictionary, so that the model can identify and suppress global redundancy more effectively.
> >
> > In contrast, MambaIRv2 is designed for image super-resolution. As a result, its architecture is not intended to detect or model redundancy for compression. This fundamental gap in task motivation naturally leads to different classification and prompting strategies.
> >
> > ---
> >
> > **Technical Comparison**
> >
> > 1. Token Classification:
> >
> >    **Our CAM: redundancy-aware clustering**
> >
> >    Our CAM block performs token classification via codebook-based KMeans clustering, which is explicitly tailored to redundancy discovery. We use cosine similarity as the distance metric to conduct KMeans over token features, updated with iterative mini-batch KMeans. This yields a shared codebook of cluster centroids. With these centroids, content-similar tokens can be grouped and reordered so that redundancy among semantically similar yet spatially distant regions is more effectively reduced.
> >
> >    ---
> >
> >
> >
> >    **MambaIRv2: unsupervised classification head**
> >
> >    MambaIRv2 instead uses a lightweight linear projection head (termed a *classification head*) to produce logits per token, followed by Gumbel-Softmax sampling for discrete assignments. There is no explicit supervision for this classification head. This routing is **not guided by any explicit semantic or redundancy-related objective**. Consequently, the resulting clusters lack clear interpretability and do not reliably reflect token redundancy.
> >
> >    ---
> >
> >
> >
> >    As shown in Fig. 19 of our revised paper, the clustering results produced by our method align well with semantic structures and visual redundancy patterns. In contrast, the classification head in MambaIRv2 yields classifications that do not correspond to semantic content or redundancy. Furthermore, Fig. 21 visualizes all 64 clusters from the classification-head strategy; none exhibits meaningful redundancy/semantic consistency. This confirms that MambaIRv2’s routing is not designed for compression, whereas our codebook-based clustering is explicitly crafted to better capture redundancy.
> >
> >
> >
> > 2. Global Prompting
> >
> >    **Our CAM: redundancy-aware dictionary**
> >
> >    We acknowledge that our prompting mechanism follows the Attentive State Space idea introduced in MambaIRv2. However, our prompts are **strictly tied to redundancy-aware clustering**.
> >    As stated in the original paper, *“Each entry in this dictionary is a prompt vector corresponding to a semantic cluster.”*
> >
> >    Specifically, with clustering centroids $[\mathbf{c}_1;\dots;\mathbf{c}_K]$, we apply a learnable linear projection $\mathcal{A}:\mathbb{R}^{d}\rightarrow\mathbb{R}^{d_s}$ to each centroid and obtain dictionary $\mathbf{U}$
> >    $$
> >    \mathbf{U} = \mathcal{A}\big([\mathbf{c}_1;\dots;\mathbf{c}_K]\big)\in\mathbb{R}^{K\times d_s},
> >    $$
> >    so that each row of $\mathbf{U}$ serves as the prompt vector projected from and associated with a cluster. Each entry carries clear semantic meaning and is tightly coupled with the clustering results.
> >
> >    We then query this dictionary using the cluster assignments. The resulting prompt signal indicates **where (in which clusters) the current features are highly redundant and where they are less redundant**. In other words, it provides a global map of redundancy distribution over semantic clusters. Clusters with heavy redundancy are explicitly highlighted for the Mamba block to focus on.
> >
> >    ---
> >
> >
> >
> >    **MambaIRv2: standalone dictionary**
> >
> >    **In contrast**, MambaIRv2 employs a prompt pool that is a standalone learnable matrix optimized directly by gradient descent **without explicit semantic constraints**. As a consequence, its prompt vectors:
> >
> >       - Do not have clearly interpretable semantic meaning or redundancy-awareness, and
> >       - Do not explicitly encode the global redundancy distribution across content clusters into state-space equation.
> >
> >
> >
> > ---
> >    **Summary of the design gap**
> >
> > In summary, our method is purpose-built for the key requirement of learned image compression: capturing and eliminating redundancy. This is why we introduce (i) a redundancy-aware clustering strategy specialized for redundancy detection, and (ii) a redundancy-aware prompt conditioning that provides the state-space model with a global redundancy distribution signal. MambaIRv2, developed for super-resolution, does not incorporate these redundancy-oriented designs. As a result, its strategy is not redundancy-aware and lacks explicit semantic awareness.

---

> ### Author Response · Authors · 2025-11-22
> **Response to Reviewer QkwG**
>
> (Continued)
> > **Experimental Evidence**
> >
> > We conduct ablations on the classification and dictionary strategies of MambaIRv2 within our framework.
> > Both qualitative and quantitative comparisons are provided to highlight the differences between our redundancy-aware token classification and cluster-aware dictionary and those of MambaIRv2.
> >
> > 1. **Classification Results.**
> >
> >    We first compare the classification strategies: our codebook-based KMeans clustering vs. the unsupervised linear classification head in MambaIRv2.
> >
> >    - As shown in Figure 19, our redundancy-aware clusters align well with semantic structures and visual redundancy patterns, whereas the assignments produced by MambaIRv2's classification head do not correspond to semantic content or redundancy.
> >    - Moreover, Figure 21 visualizes all 64 clusters produced by the classification-head strategy; none exhibits meaningful redundancy or semantic consistency.
> >
> >    This indicates that MambaIRv2's routing is not designed for compression, while our clustering is explicitly tailored to capture redundancy.
> >
> > 2. **Single-image ERF visualization.**
> >
> >    Figure 20 in the revision further provides single-image ERF visualizations.
> >
> >    - Our ERF concentrates sharply on redundant, content-correlated tokens and regions, including spatially distant areas, demonstrating strong content adaptivity.
> >    - In contrast, MambaIRv2 exhibits a more content-agnostic long-range pattern: the responses are relatively uniformly spread over the entire image, with much weaker sensitivity to semantic structure and redundancy.
> >
> >    This shows that our CAM captures redundancy more effectively, which in turn leads to improved compression performance.
> >
> > 3. **BD-rate comparison.**
> >
> >    We implement two variants to quantitatively compare the classification and prompt-pool strategies within our compression framework.
> >
> >    - The first variant, "CMIC (w/ Standalone Dictionary)", replaces our cluster-aware dictionary with the standalone dictionary in MambaIRv2.
> >
> >    - The second variant, "CMIC (w/ Classification Head)", replaces our redundancy-aware clustering with the unsupervised classification head from MambaIRv2.
> >
> >      Since our cluster-aware dictionary relies on the clustering centers, for "CMIC (w/ Classification Head)" we have to use the standalone dictionary for prompt conditioning.
> >
> >    The resulting BD-rate comparisons show that our CAM design consistently outperforms the MambaIRv2-style variant, confirming the benefit of our redundancy-aware classification and dictionary.
> >
> > | Model                           | Kodak  | Tecnick | CLIC   |
> > | ------------------------------- | ------ | ------- | ------ |
> > | CMIC (CAM)                      | -15.91 | -21.34  | -17.58 |
> > | CMIC (w/ Standalone Dictionary) | -15.02 | -20.21  | -16.73 |
> > | CMIC (w/ Classification Head)   | -14.46 | -19.76  | -15.83 |

---

> ### Author Response · Authors · 2025-11-22
> **Response to Reviewer QkwG**
>
> ### 📝 Q2: "The comparison with the latest approaches, MLICv2"
>
> **Response:**
>
> Thank you for your suggestion. We provide the comparisons here.
>
> |             | Kodak  | Tecnick | CLIC   | Params | TFLOPs | Dec.Lat. | Peak Mem |
> | ----------- | ------ | ------- | ------ | ------ | ------ | -------- | -------- |
> | MLICv2      | -16.16 | -20.13  | -15.79 | 84.30  | 2.78   | 0.520    | 5.07     |
> | CMIC (Ours) | -15.91 | -21.34  | -17.58 | 69.11  | 2.39   | 0.405    | 4.44     |
>
> *Note:* MLICv2 is not open-sourced; all numbers are extracted from their paper.
>
> 1. Compared with MLICv2, our CMIC uses about 20% fewer parameters, and also incurs lower TFLOPs, decoding latency, and peak memory.
> 2. In terms of RD performance, MLICv2 is marginally better on Kodak, while CMIC clearly outperforms MLICv2 on the higher-resolution Tecnick and CLIC2020 datasets. We attribute this advantage to our CAM block: unlike MLICv2, which mainly strengthens the entropy model, CMIC focuses on improving the transform networks. The stronger gains on high-resolution datasets highlight the effectiveness of CAM in capturing **global, spatially-distant redundancy**, which becomes more pronounced as image resolution increases.
>
> We further emphasize that MLICv2 and CMIC target largely orthogonal components: MLICv2 improves the entropy model, whereas CMIC improves the transform side. Therefore, their strengths are complementary, and combining an advanced entropy model with our CAM-based transform networks is a promising direction for further RD gains.
>
> Although MLICv2 itself is not released, we conduct an additional experiment by plugging the entropy model of MLIC++ into CMIC at λ=0.05. The results show that a more advanced entropy model can further boost CMIC, confirming the complementarity between the two lines of improvement. We will complete training of other λ values and provide the full BD-rate results very soon.
>
> | Model                     | Loss (λ=0.05) |
> | ------------------------- | ------------: |
> | CMIC                      |         1.354 |
> | CMIC+EntropyModel(MLIC++) |         1.339 |

---

> ### Author Response · Authors · 2025-11-28
> **Additional Response to Reviewer QkwG**
>
> (Continued response to Q2: "The comparison with the latest approaches, MLICv2")
>
> | Model                       | Kodak  | Tecnick | CLIC   |
> | :-------------------------- | :----- | :------ | :----- |
> | CMIC                        | -15.91 | -21.34  | -17.58 |
> | MLICv2                      | -16.16 | -20.13  | -15.79 |
> | CMIC + EntropyModel (MLIC++)| -17.43 | -21.84  | -18.21 |
>
> We have finished the ablation study on the entropy model by replacing our default entropy model with that of MLIC++, yielding the variant "CMIC + EntropyModel (MLIC++)." As shown in the table, this stronger entropy model consistently improves CMIC across all three benchmarks. Moreover, with the relatively older (but still advanced) entropy model of MLIC++, our CMIC variant already outperforms MLICv2 on all three datasets, highlighting the superiority of our transform network design.

---

> ### Author Response · Authors · 2025-11-28
> **Kind Follow-up Regarding Reviewer's Remaining Questions**
>
> We thank the reviewers again for their careful reading of our paper and thoughtful comments.
>
> As the discussion deadline is approaching, we would like to kindly confirm whether our rebuttal and revisions have sufficiently addressed your concerns. We would be pleased to provide further clarifications or conduct additional experiments if there are any remaining questions.

---

### Official Review · Reviewer_GLne · 2025-10-30

**Soundness:** 3
**Presentation:** 3
**Contribution:** 2
**Rating:** 4
**Confidence:** 5

**Summary:**

This paper proposes a clustering based content based scanning method and a global prompt based non causal modeling approach to address the issues of content independent scanning order and causal characteristics in the Mamba structured learned image codec. Ultimately, while preserving the original linear complexity of Mamba, this approach is effective, ,  reducing BD-rate by 15.91%, 21.34%, and 17.58% on the Kodak, Tecnick, and CLIC datasets over VTM-21.0.

**Strengths:**

- Good RD performance while maintaining low complexity
- Relieve the high memory usage problem of prior Mamba based learned image codec

**Weaknesses:**

- The methods proposed for both clustering partitioning and global prior modulation are common in prior "content-adaptive" studies, such as Rounting Transformer [1], even in the field of learned image compression, there are similar jobs available [2].
- The main contribution of this paper is to improve the Mamba module in the Mamba-based learned image codec. However, it seems that there are many other differences compared to the previous works, e.g., MambaVC and MambaIC, including the entropy models, etc. The paper lacks detailed performance analysis.
- Regarding the difference in receptive fields, according to Appendix Fig.16, it seems that this global receptive field and the so-called content aware features are also present in MambaIC, What is the actural difference?
- The paper lacks a performance comparison with the latest methods listed, e.g., [3].

Ref:

[1] Roy A, Saffar M, Vaswani A, et al. Efficient content-based sparse attention with routing transformers[J]. Transactions of the Association for Computational Linguistics, 2021, 9: 53-68.

[2] Zhang Y, Duan Z, Lu M, et al. Another way to the top: Exploit contextual clustering in learned image coding[C]//Proceedings of the AAAI Conference on Artificial Intelligence. 2024, 38(8): 9377-9386.

[3] Li Y, Zhang H, Li L, et al. Learned image compression with hierarchical progressive context modeling[C]//Proceedings of the IEEE/CVF International Conference on Computer Vision. 2025: 18834-18843.

**Questions:**

- I can understand that the method proposed in this paper can improve the memory consumption of Mamba based image codecs, but further analysis is needed to determine the source of the model's performance improvement, including comparisons with existing Mamba-based methods such as MambaIC, and the benefits of visualizing/analyzing the proposed non causal approach.

---

> ### Author Response · Authors · 2025-11-22
> **Response to Reviewer GLne**
>
> ### 📝 Q1: "There are "content-adaptive" studies available, such as Routing Transformer[1] and [2] "
>
> **Response:**
>
> We agree that prior work has explored content-adaptive ideas. We added corresponding citation and discussion with  routing transformer [1] in Section 2.3 CONTENT-ADAPTIVE IMAGE COMPRESSION. The method [2] brings this idea into CNN-based learned image compression, where clustering is used to enhance the grouping of convolutional features. We provide a detailed discussion of the relationship between our method and [1] and demonstrate our superiority in Appendix A.2.
>
> In contrast, our work focuses on making Mamba content-adaptive while preserving its linear-time global modeling. Mamba already offers strong global context modeling and is widely regarded as a promising alternative to Transformers. For image compression, however, the core challenge is different from the Transformer complexity-globality trade-off.
>
> **Our key insight is that, content-agnostic and purely raster-based scanning of Mamba poorly matches the redundancy structure among tokens.**
>
> Our CAM design directly targets this issue by introducing a content-adaptive mechanism that encourages interactions between semantically similar tokens, and preserves Mamba’s linear complexity. This is the key motivation and contribution that distinguishes our method from existing content-adaptive approaches such as Routing Transformer and [1].
>
>
>
> ---
>
>
>
> ### 📝 Q2: "There are many other differences compared to the previous works, e.g., MambaVC and MambaIC, including the entropy models, etc."
>
> **Response:**
>
> Thank you for your suggestion. We choose the stronger model MambaIC as our reference and align the entropy model accordingly. Specifically, MambaIC adopts an advanced hybrid autoregressive entropy model that combines autoregressive context modeling with window-based local attention.
>
> We have so far completed training at λ=0.05 and λ=0.025; the corresponding test losses on Kodak are:
>
> | Model                      | Loss (λ=0.05) | Loss (λ=0.025) |
> | -------------------------- | ------------: | -------------: |
> | MambaIC                    |         1.385 |          1.032 |
> | CMIC                       |         1.354 |          1.010 |
> | CMIC+EntropyModel(MambaIC) |         1.344 |          0.999 |
>
> We can see that equipping CMIC with the advanced entropy model of MambaIC further improves performance, which demonstrates that our CAM-based transform network remains effective and complementary to stronger entropy models. We will complete training for other λ values and report full BD-rate results here very soon.
>
> ---
>
> More importantly, to directly demonstrate the effectiveness of our CAM block (rather than only the entropy model), we have replaced CAM with 1D and 2D Mamba in our ablation studies:
>
> - As shown in Table 2, when CAM is replaced by 1D Mamba, the RD performance degrades by 2.7%–3.6%, especially on high-resolution datasets.
> - As shown in Table 4, our CAM outperforms 2D Mamba even with a single scan, while 2D Mamba requires four directional scans. Note that 2D Mamba is exactly the basic transform block used in MambaVC and MambaIC.
>
> These results clearly indicate that the performance gains mainly come from our CAM-based transform networks.

---

> ### Author Response · Authors · 2025-11-22
> **Response to Reviewer GLne**
>
> ### 📝 Q3: "It seems that this global receptive field and the so-called content aware features are also present in MambaIC"
>
> **Response:**
>
> We agree that MambaIC has a global receptive field, but its effective receptive field (ERF) is not truly content-adaptive.
>
> In short, although both methods have global ERF support, MambaIC produces a content-agnostic cross pattern driven by fixed four-directional scanning and thus misaligns with real image redundancy, whereas CMIC breaks the Euclidean neighbor bias to aggregate truly semantic, redundant regions via non-local, content-adaptive context selection.
>
> In detail:
> > As shown in Fig. 16:
> >
> > 1. **Globality vs. adaptivity.** Both CMIC and MambaIC exhibit certain global support in their ERFs (MambaIC uses four-directional global scanning, which indeed captures long-range information). However, the key difference lies in how this global context is selected.
> >
> > 2. **MambaIC exhibits a content-agnostic “cross” pattern.** Across almost all examples, MambaIC’s ERF shows a very stable cross-shaped pattern: given a current token, the entire horizontal row and vertical column are strongly activated. This indicates that the model systematically aggregates context along the four scan directions, largely dictated by its four-way sequential scanning mechanism rather than by the actual image content. In other words, this is a **content-agnostic context pattern** induced by raster-scan inductive bias.
> >
> > 3. **This cross pattern is misaligned with redundancy in real images.** In Kodak Kodim17, for instance, it is visually obvious that redundancy is mainly concentrated around the statue’s head. Yet MambaIC’s ERF extends vertically across the whole column into the background stone wall, and horizontally through many regions that are almost irrelevant to the current block. These positions are not genuinely “highly redundant and strongly correlated,” but they still receive systematically high attention due to the fixed scan directions.
> >
> > 4. **CMIC's ERF follows semantic structure and redundancy.** In contrast, CMIC's ERF is much better aligned with semantic content. On Kodim17, CMIC's responses concentrate around the head contour; on Kodim16, they focus on the clouds and island areas. These are precisely the regions that are both highly correlated with the current token and perceptually redundant. CMIC rarely exhibits entire-row or entire-column patterns, suggesting that its context selection is driven by image content rather than a fixed scanning path.
> >
> > 5. **Underlying reason: Euclidean neighbor bias vs. non-local content search.** This difference stems from the scanning behavior of MambaIC vs. CMIC:
> >
> >    MambaIC’s four-directional raster scan still imposes a strong Euclidean neighbor bias: tokens with the highest responses along horizontal and vertical directions tend to be those that are nearest in the 4-direction scan order.
> >
> >    CMIC, on the other hand, breaks this Euclidean constraint and explicitly searches for long-range, non-local dependencies. It focuses on regions that are truly semantically related and redundant with the current token, demonstrating stronger content adaptivity.
> >
> > In summary, simply having a global receptive field does not guarantee content adaptivity. Our visualizations show that CMIC performs genuinely content-driven, redundancy-aware context aggregation, while MambaIC’s ERF is largely governed by its fixed four-directional scanning prior. This also demonstrates the effectiveness of our adaptive strategies.
>
>
>
> ---
>
>
>
> ### 📝 Q4: "Comparison with the latest method HPCM"
>
> **Response:**
>
> Thank you for the suggestion. We add the comparison below:
>
> |             | Kodak  | Tecnick | CLIC   | Params | TFLOPs | Dec.Lat. | Peak Mem |
> | ----------- | ------ | ------- | ------ | ------ | ------ | -------- | -------- |
> | HPCM        | -14.33 | -16.77  | -14.54 | 68.49  | 2.21   | 0.304    | 2.97     |
> | CMIC (Ours) | -15.91 | -21.34  | -17.58 | 69.11  | 2.39   | 0.405    | 4.44     |
>
> HPCM is an efficient model released on 25 Jul 2025, which is concurrent with our work.
>
> Our model consistently outperforms HPCM on all three datasets. The performance gap is more pronounced on the higher-resolution datasets (Tecnick and CLIC), which highlights the advantage of our CAM block in capturing long-range redundancy.
>
> We also emphasize that HPCM mainly focuses on improving the entropy model, while our work focuses on strengthening the transform network. These two directions are largely orthogonal, suggesting that future work could further boost RD performance by combining the strengths of both approaches.

---

> ### Author Response · Authors · 2025-11-22
> **Response to Reviewer GLne**
>
> ### 📝 Q5: "Visualizing/analyzing the proposed non causal approach."
>
> **Response:**
>
> Thank you for your suggestion. We compute the ERF between the input and output features of a single state-space layer with soft clustering to analyze the proposed non-causal mechanism and to isolate the effects of our two main components: CTP (Content-Adaptive Token Permutation) and GPP (Global-Prior Prompting).
>
> 1. As shown in Fig. 9 in the revised paper, when both CTP and GPP are removed (column (b)), the tokens are processed in a standard raster-scan order. Taking the center token as the anchor, we observe that the ERF stops exactly at the central position (H//2,W//2): even the subsequent tokens in the same row exhibit *zero* ERF values. This reflects strict raster-scan causality, where each token is conditioned only on the previously scanned tokens.
>
> 2. In column (c), when we enable GPP, we observe non-zero activations even after the scanned sequence. This shows that GPP allows the state-space model to “see beyond” the strictly causal scan and gain stronger global semantic awareness. Moreover, the activated regions are semantically aligned and meaningful, indicating that the learned prompt encodes effective global semantic conditions that guide the Mamba scan.
>
> 3. In columns (d) and (e), once CTP is applied, the ERF maps no longer exhibit the characteristic raster-scan pattern. Instead, activation spreads over semantically related locations. This demonstrates that CTP effectively breaks the fixed Euclidean-neighbor constraint imposed by pre-defined scan orders, prioritizing feature-space proximity over spatial adjacency and further enhancing redundancy elimination between distant yet content-correlated tokens.
>
>
> **Corresponding discussions have been added to Line 473-505 in the main paper.**
>
> ---
>
> [1] Roy, Aurko, et al. "Efficient content-based sparse attention with routing transformers." *Transactions of the Association for Computational Linguistics* 9 (2021): 53-68.
>
> [2] Zhang, Yichi, et al. "Another way to the top: Exploit contextual clustering in learned image coding." *Proceedings of the AAAI Conference on Artificial Intelligence*. Vol. 38. No. 8. 2024.

---

> ### Author Response · Authors · 2025-11-27
> **Additional Response to Reviewer GLne**
>
> (Continued response to Q2 “There are many other differences compared to the previous works, e.g., MambaVC and MambaIC, including the entropy models, etc.”).
>
>
> | Model                       | Kodak  | Tecnick | CLIC   |
> | :-------------------------- | :----- | :------ | :----- |
> | CMIC                        | -15.91 | -21.34  | -17.58 |
> | MambaIC                     | -13.01 | -15.27  | -15.23 |
> | CMIC + EntropyModel (MambaIC) | -16.97 | -21.45  | -17.89 |
>
> We have completed the ablation study on the entropy model by replacing our default entropy model with that of MambaIC, yielding the variant "CMIC + EntropyModel (MambaIC)". As shown in the table, this stronger entropy model consistently improves CMIC across all three benchmarks, with the largest gain on the Kodak dataset. Moreover, compared to MambaIC, our CMIC variant with its entropy model significantly outperforms MambaIC on all three datasets, demonstrating the effectiveness of our transform network over the 2D Mamba-based transform network.
>
> Please note that this variant ("CMIC + EntropyModel (MambaIC)") contains 159.75M parameters, which is substantially larger than the original CMIC model (69.11M).

---

> ### Author Response · Authors · 2025-11-28
> **Kind Follow-up Regarding Reviewer's Remaining Questions**
>
> We thank the reviewers again for their careful reading of our paper and thoughtful comments.
>
> As the discussion deadline is approaching, we would like to kindly confirm whether our rebuttal and revisions have sufficiently addressed your concerns. We would be pleased to provide further clarifications or conduct additional experiments if there are any remaining questions.

---

> > ### Comment · Reviewer_GLne · 2025-11-28
> >
> > Thank you for your response. Most of my concerns have been addressed, and I also encourage the authors to release the code, which would greatly enhance reproducibility and benefit the community.
> >
> > However, due to a current OpenReview bug, score updating is not yet enabled on the reviewers’ side. Once it becomes available, I will revise my score accordingly.

---

> > > ### Author Response · Authors · 2025-11-28
> > >
> > > Thank you very much for your encouraging feedback and for your willingness to update the score. We will release our code and checkpoints immediately upon acceptance to support reproducibility.

---

### Official Review · Reviewer_72TK · 2025-10-30

**Soundness:** 3
**Presentation:** 3
**Contribution:** 2
**Rating:** 4
**Confidence:** 5

**Summary:**

This paper presents a mamba-based image compression method that enhances mamba’s adaptability to the input content. In mamba, the predefined raster (or multi-directional) scans are content-independent, which limits the effective elimination of redundancy between tokens that are content-correlated but spatially distant. To address this, the authors introduce a content-adaptive token permutation strategy to enhance interactions between similar tokens. Additionally, they adopt strategies from mambairv2, such as injecting sample-specific global priors into the state-space model. Experiments demonstrate the method's effectiveness, surpassing VTM-21.0 by 15.91%, 21.34%, and 17.58% in BD-rate on the Kodak, Tecnick, and CLIC datasets, respectively.

**Strengths:**

1.	The writing quality is good, and the explanation is clear.
2.	The motivation for improving mamba’s fixed scanning order is reasonable, and enhancing adaptability is a valid direction.
3.	The paper provides a comprehensive evaluation of complexity, including model size, FLOPs, latency, and memory usage.
4.	The ERF visualizations clearly demonstrate the model's global perception ability and content adaptiveness.

**Weaknesses:**

5. The paper compares only with MambaIC from CVPR 2025, ignoring LALIC and DCAE. Considering that LALIC and DCAE released code several months ago, a comparison with these methods is essential.
6. The transform module in CMIC appears to be much deeper than in other methods. In Stage 2 and Stage 3, there are four blocks of attention and CAM in total—is this correct?
7. The proposed Learnable Prompt Dictionary is essentially the Attentive State Space Module from mambairv2. The authors do not mention this in the paper, but since they cite mambairv2, they must be aware of it. I believe this is problematic and undermines the paper’s contribution and originality.
8. In the supplementary materials, the authors compare their method with Zhang et al.’s approach. Are there quantitative results that align with other modules?
9. Based on Figure 1, it seems that the clustering only scans within the current class. Does this harm the global nature of the method?
10. In the ablation study, using just the 2D mamba outperforms MambaIc—what is the reason for this?
11. According to Tables 2 and 3, removing CTP and GPP results in worse performance than just using 2D mamba. However, using pure mamba + CTP + GPP (CAM) does not match CMIC. What is the reason for this inconsistency?

**Questions:**

Please refer to weaknesses

---

> ### Author Response · Authors · 2025-11-22
> **Response to Reviewer 72TK**
>
> ### 📝 Q1: "The paper compares only with MambaIC from CVPR 2025, ignoring LALIC and DCAE."
>
> **Response:**
>
> We thank the reviewer for pointing this out. In the revised version, we add explicit comparisons with LALIC [1] and DCAE [2]. The table below summarizes the BD-rate and complexity metrics:
>
> |             | Kodak  | Tecnick | CLIC   | Params | TFLOPs | Dec.Lat. | Peak Mem |
> | ----------- | ------ | ------- | ------ | ------ | ------ | -------- | -------- |
> | DCAE        | -15.40 | -19.62  | -16.46 | 119.22 | 2.28   | 0.478    | 5.59     |
> | LALIC       | -13.68 | -17.26  | -15.01 | 63.24  | 2.53   | 0.385    | 4.09     |
> | CMIC (Ours) | -15.91 | -21.34  | -17.58 | 69.11  | 2.39   | 0.405    | 4.44     |
>
> - Our model consistently outperforms LALIC and DCAE across all three datasets while maintaining comparable computational complexity. Moreover, it achieves these gains with 58% of the parameter count of DCAE.
>
> - The performance gap in favor of our method is larger on the higher-resolution datasets (Tecnick and CLIC), highlighting the advantage of our CAM block in capturing global redundancy.
>
>
>
>
> ---
>
> ### 📝 Q2: "The transform module in CMIC appears to be much deeper than in other methods."
>
> **Response:**
>
> In Stages 2 and 3, our transform module contains two Attention blocks and two CAM blocks per stage (4 blocks per stage in total). However, this does **not** imply that CMIC is deeper or more complex overall.
>
> To make the comparison clearer, we summarize the depth of each transform stage and the overall complexity:
>
> | Method      | Depth per transform stage | Params | TFLOPs |
> | ----------- | ------------------------- | ------ | ------ |
> | DCAE [2]    | [1, 2, 12]                | 119.22 | 2.28   |
> | LALIC [1]   | [2, 4, 6]                 | 63.24  | 2.53   |
> | MambaIC [3] | [2, 2, 9]                 | 157.09 | 5.56   |
> | CMIC (Ours) | [3, 4, 4]                 | 69.11  | 2.39   |
>
> Different SOTA methods adopt different architectural designs and stage configurations, so precisely “aligning” depth across them is inherently difficult. Instead, it is easier to compare overall complexity, such as parameter count and FLOPs. From the table, we can see that:
>
> - CMIC has a moderate depth with much fewer parameters than DCAE and MambaIC, and lower TFLOPs than LALIC and MambaIC.
> - In particular, compared with the SOTA Mamba-based method MambaIC, our model uses **less than half the parameters** and **less than half the TFLOPs**, while achieving superior RD performance.

---

> ### Author Response · Authors · 2025-11-22
> **Response to Reviewer 72TK**
>
> ### 📝 Q3: Comparison with prompting mechanism in MambaIRv2
>
> **Response:**
>
> Thank you for the comment. Yes, the Attentive State-Space Equation in our method indeed follows MambaIRv2 [4], and we will revise the paper to state this explicitly.
>
> However, we emphasize that our prompt dictionary is fundamentally different from that of MambaIRv2.
>
> In short, we introduce a **redundancy-aware dictionary** specifically designed for redundancy elimination in image compression. It is explicitly coupled with the clustering results, so the resulting prompts reflect the underlying redundancy distribution. In contrast, MambaIRv2 uses a **standalone dictionary** learned without such coupling, and thus lacks semantic interpretability and redundancy awareness.
>
>
> In detail:
> > **CMIC:**
> >
> > Our core motivation is to enhance Mamba’s ability to eliminate redundancy between tokens that are content-correlated but spatially distant. To this end, CAM employs redundancy-aware clustering and redundancy-aware dictionary:
> >
> > - **Step 1 – Redundancy-aware Clustering.**
> >   We first use cosine similarity as the distance metric to perform K-means clustering over tokens, with iterative mini-batch K-means updates. This yields a shared codebook of cluster centroids. Based on these centroids, content-similar tokens can be reordered and grouped together so that redundancy across content-similar but spatially distant regions can be better eliminated.
> >
> > - **Step 2 – Redundancy-aware Dictionary.**
> >   As stated in the original paper, *“Each entry in this dictionary is a prompt vector corresponding to a semantic cluster.”*
> >
> >   Specifically, with clustering centroids $[\mathbf{c}_1;\dots;\mathbf{c}_K]$, we apply a learnable linear projection $\mathcal{A}:\mathbb{R}^{d}\rightarrow\mathbb{R}^{d_s}$ to each centroid and obtain dictionary $\mathbf{U}$
> >   $$
> >   \mathbf{U} = \mathcal{A}\big([\mathbf{c}_1;\dots;\mathbf{c}_K]\big)\in\mathbb{R}^{K\times d_s},
> >   $$
> >   so that each row of $\mathbf{U}$ serves as the prompt vector projected from and associated with a cluster. Each entry carries clear semantic meaning and is tightly coupled with the clustering in Step 1.
> >
> >   We then query this dictionary using the cluster assignments obtained in Step 1. The resulting prompt signal indicates **where (in which clusters) the current features are highly redundant and where they are less redundant**. In other words, it provides a global map of redundancy distribution over semantic clusters. Clusters with heavy redundancy are explicitly highlighted for the Mamba block to focus on.
> >
> > ---
> >
> > **MambaIRv2:**
> >
> > **In contrast**, MambaIRv2 employs a prompt pool that is a standalone learnable matrix optimized directly by gradient descent **without explicit semantic constraints**. As a consequence, its prompt vectors:
> >
> > - Do not have clearly interpretable semantic meaning or redundancy-awareness, and
> > - Do not explicitly encode the global redundancy distribution across content clusters into state-space equation.
> >
> > ---
> >
> > **Quantitative Comparison:**
> >
> > We also provide quantitative results ablating this dictionary strategy. "CMIC (w/ Standalone Dictionary)" replaces our redundancy-aware dictionary with the standalone dictionary in MambaIRv2. The resulting BD-rate comparisons show that our CAM design consistently outperforms the MambaIRv2-style variant, confirming the benefit of our redundancy-aware classification and dictionary.
> >
> > | Model                           | Kodak  | Tecnick | CLIC   |
> > | ------------------------------- | ------ | ------- | ------ |
> > | CMIC (CAM)                      | -15.91 | -21.34  | -17.58 |
> > | CMIC (w/ Standalone Dictionary) | -15.02 | -20.21  | -16.73 |
> >
> > ---
> >
> > In conclusion, while our method is inspired by the idea of making selective scanning non-causal via prompting, it substantially extends and adapts this idea to the image compression, explicitly targeting redundancy elimination.
> >
> >
>
> We appreciate the reviewer’s suggestion and add more details to clarify these differences more clearly in Line 265-307 in the main paper.

---

> ### Author Response · Authors · 2025-11-22
> **Response to Reviewer 72TK**
>
> ### 📝 Q4: The authors compare their method with Zhang et al.’s approach. Are there quantitative results that align with other modules?"
>
> **Response:**
>
> Thank you for the suggestion. To align with Zhang et al. [5], we replace our CAM block with the CCB block used in [5], keeping all other components unchanged.
>
> | Method     |   Kodak | Tecnick | CLIC2020 |
> | ---------- | ------: | ------: | -------: |
> | CCB        |  -9.92% | -10.67% |   -8.73% |
> | CAM (Ours) | -15.91% | -21.34% |  -17.58% |
>
> The CAM variant significantly outperforms the CCB variant, especially on the high-resolution Tecnick and CLIC2020 datasets. This demonstrates that CAM has a much stronger ability to capture global redundancy than CCB.
>
> ---
>
> ### 📝 Q5: "Based on Figure 1, it seems that the clustering only scans within the current class."
>
> **Response:**
>
> Sorry for the misunderstanding. After clustering and reordering the tokens, we apply 1D Mamba to the **entire reordered sequence**, not to each class independently.
>
> - Tokens from different classes are concatenated and scanned sequentially according to the order of cluster centers.
> - Therefore, Mamba still operates on a globally ordered sequence and can propagate information across all clusters.
>
> In other words, clustering and reordering improve the local coherence of the sequence (content-similar tokens are placed closer), while preserving the global nature of Mamba’s scanning over the full sequence.
>
> ---
>
> ### 📝 Q6: "Using just the 2D mamba outperforms MambaIC"
>
> **Response:**
>
> In our ablation study (Table 3), the variant that “uses 2D Mamba” is constructed by replacing the CAM block with 2D Mamba, while keeping the interleaved Attention blocks unchanged. This setting is designed specifically to isolate the effectiveness of CAM against 2D Mamba.
>
> Thus:
>
> - The ablation model is not equivalent to MambaIC, since it still contains interleaved Attention blocks, which are crucial for capturing local redundancy.
> - The combination of 2D Mamba + Attention in our architecture is effective, and this is why this ablation variant can slightly outperform MambaIC.
>
> We apologize for the confusion and clarify this ablation configuration more explicitly in the revised paper.
>
>
>
>
> ---
>
>
> ### 📝 Q7: "Removing CTP and GPP results in worse performance than just using 2D Mamba"
>
> **Response:**
>
> Removing CTP and GPP effectively degrades the CAM block to a **vanilla 1D Mamba** block, which, as the reviewer notes, indeed performs worse than 2D Mamba.
>
> - **1D Mamba** scans the sequence only once along the predefined raster order.
> - **2D Mamba** in MambaIC scans four times along four directions, effectively increasing the number of scanning passes and improving its ability to capture context.
>
> This observation directly motivates our CAM design: instead of simply increasing the number of scanning directions, we prioritize interactions between spatially distant but content-correlated tokens through semantic clustering and prompt conditioning.
>
>
> ---
>
>
> ### 📝 Q8: "Using pure mamba + CTP + GPP (CAM) does not match CMIC."
>
> **Response:**
>
> Thank you for raising the question. But we are not entirely sure about the reviewer’s concern. Could you please clarify the specific issue?
>
>
>
> ---
>
> [1] Feng, Donghui, et al. "Linear Attention Modeling for Learned Image Compression." *Proceedings of the Computer Vision and Pattern Recognition Conference*. 2025.
>
> [2] Lu, Jingbo, et al. "Learned Image Compression with Dictionary-based Entropy Model." *Proceedings of the Computer Vision and Pattern Recognition Conference*. 2025.
>
> [3] Zeng, Fanhu, et al. "MambaIC: State Space Models for High-Performance Learned Image Compression." *Proceedings of the Computer Vision and Pattern Recognition Conference*. 2025.
>
> [4] Guo, Hang, et al. "Mambairv2: Attentive state space restoration." *Proceedings of the Computer Vision and Pattern Recognition Conference*. 2025.
>
> [5] Zhang, Yichi, et al. "Another way to the top: Exploit contextual clustering in learned image coding." *Proceedings of the AAAI Conference on Artificial Intelligence*. Vol. 38. No. 8. 2024.

---

> ### Author Response · Authors · 2025-11-28
> **Kind Follow-up Regarding Reviewer's Remaining Questions**
>
> We thank the reviewers again for their careful reading of our paper and thoughtful comments.
>
> As the discussion deadline is approaching, we would like to kindly confirm whether our rebuttal and revisions have sufficiently addressed your concerns. We would be pleased to provide further clarifications or conduct additional experiments if there are any remaining questions.

---

### Author Response · Authors · 2025-11-22
**General Responses to All Reviewers**

We sincerely thank all reviewers for their constructive feedback. We have updated the manuscript, and all revisions are highlighted in blue. We summarize several key clarifications and substantive additions as follows:

1. **Broader performance comparisons.**
    We have added explicit BD-rate/complexity comparisons with previously missing and latest baselines (LALIC, DCAE, HPCM, MLICv2). CMIC achieves better or comparable RD performance under similar or lower Params/TFLOPs, with more pronounced gains on high-resolution datasets, highlighting CAM’s strength in capturing global redundancy.

2. **Comparison with MambaIRv2.**
    We clarify that CAM is fundamentally different from MambaIRv2. We propose redundancy-aware clustering plus redundancy-aware dictionary, which is purpose-built for compression rather than super-resolution.

   We have provided ablations with the MambaIRv2 strategy and show that our design is superior in terms of both BD-rate and qualitative results (clustering visualizations and ERF analysis).

3. **Additional visualizations for content adaptivity and non-causality.**
    We have added single-layer ERF analysis to demonstrate the effects of CTP and GPP. The visualizations show that CMIC’s ERF focuses on semantically redundant regions and effectively breaks the raster-scan causality.

4. **Complementarity with improved entropy models.**
    We claim that our work focuses on leveraging Mamba to design more advanced transform networks that better eliminate global redundancy. We note that some recent studies improve entropy models, which is largely orthogonal to our contribution. We have further demonstrated that a stronger entropy model from MLIC++/MambaIC can be combined with CMIC to further boost its performance.

We would be pleased to provide further clarifications or additional analyses if the reviewers have any remaining questions.

---

### Comment · Area_Chair_URSM · 2025-11-25

Dear Reviewers,

Thank you for your time and effort in reviewing submissions for ICLR 2026. As we begin the author-reviewer discussion process, we kindly remind you to submit your responses to the author rebuttals by **December 2**.

Your engagement in this discussion phase is crucial to ensuring a fair and thorough evaluation of each submission.

### **Action Required**
- Carefully consider the authors’ rebuttal and any additional evidence they provide.
- Update your review (if applicable) to reflect your revised perspective.
- Discuss with the authors if further details are required

Your AC

---

### Author Response · Authors · 2025-12-01
**Rebuttal Summary and Clarification of Reviewer QkwG's Factual Misunderstanding**

Dear Area Chair,

We summarize the rebuttal outcome in the table bellow.

In particular, we have to report that reviewer QkwG raises only two weaknesses&questions, and one of them is based on a **`factual misunderstanding of our method`**. We submitted a detailed rebuttal well before **`22 Nov 2025`** and later posted a polite reminder, but we have not received any reply from QkwG.

We sincerely ask the Area Chair to carefully consider our clarifications and additional results in the rebuttal and revision when interpreting reviewer QkwG's rating and forming the final recommendation.

| Reviewer | Overall stance                 | Follow-up status                     | Main concerns                                                | Our response / clarification                                 |
| :------: | :----------------------------- | :----------------------------------- | :----------------------------------------------------------- | :----------------------------------------------------------- |
| **72TK** | Unclear (no updated score)     | ⚠️ **No reply** after rebuttal + ping | Details of ablation settings and model designs               | **Fully addressed point-by-point.** Detailed clarifications and additional results. |
| **GLne** | **Positive; score increased**  | ✅ Confirmed concerns addressed       | More performance analysis and additional experiments         | More visualizations and further experiments.                 |
| **QkwG** | Unclear (no updated score)     | ⚠️ **No reply** after rebuttal + ping | **Only two concerns in total `(one factual misunderstanding)`:** (1) **Severe misunderstanding** that our method ≈ MambaIRv2; (2) Comparisons with recent SOTA | **Fully addressed point-by-point.** Added extensive **visualizations and quantitative & qualitative comparisons** vs MambaIRv2, showing our method is distinct and specifically tailored for compression. |
| **ysic** | **Positive; score maintained** | ✅ Confirmed concerns addressed       | Clarifications on methodology and suggestions on paper structure | Detailed clarifications and corresponding adjustments to paper. |

---

> ### Author Response · Authors · 2025-12-01
> **Further Details**
>
> Across all four reviews, our work is consistently recognized for its **clear writing, strong motivation, and well-justified design**. Reviewers praise our comprehensive complexity analysis, noting that we achieve good RD results with low complexity and effectively alleviate the high memory usage of prior Mamba-based codecs. They also find our visualizations and figures clear and informative, demonstrating the model’s global perception ability and content adaptiveness on standard benchmarks.
>
> During the rebuttal:
>
> - **Coverage of all concerns.**
>   In the rebuttal and revised manuscript, we responded point-by-point to **all** weaknesses and questions raised by the four reviewers, including additional analyses, visualizations, new experiments, and clarifications where needed.
>
> - **Reviewers acknowledging concerns are addressed.**
>   Two reviewers, **GLne** and **ysic**, explicitly stated that their concerns have been addressed by our rebuttal and revision. Reviewer **GLne** indicated an intention to **raise the score**, and reviewer **ysic** decided to maintain their already positive assessment.
>
> - **Lack of follow-up from the remaining two reviewers.**
>   The other two reviewers, **72TK** and **QkwG**, **did not provide any follow-up after we posted our rebuttal well before 22 Nov 2025** and later politely asked whether they had any remaining concerns. Their original comments therefore do not reflect the new material we have provided.
>
> - **Reviewer-specific clarifications**
>   - **Reviewer QkwG.**
>     Reviewer QkwG raised **only two weaknesses/questions**.
>     1. **Factual misunderstanding of our method.** The first concern is based on a clear factual error: they assumed that our method is essentially identical to MambaIRv2. In response, both in the revision and rebuttal, we added detailed **quantitative and qualitative comparisons** from multiple angles, including motivation, core methodology, visualizations, ablation studies, and RD performance. These additions clearly demonstrate that our approach is **fundamentally different from MambaIRv2** and is specifically tailored for compression.
>     2. **Comparison with recent SOTA.** For the second concern regarding comparisons to recent SOTA methods, we added comprehensive experimental results against **strong recent baselines**, together with further analyses, directly addressing this request.
>
>   - **Reviewer 72TK.**
>     Reviewer 72TK mainly raised issues about the **ablation settings** and some **methodological details**. We addressed these concerns **point-by-point** by detailing the ablation settings, clarifying architectural and method details, and adding further explanations in the main text and appendix.
>
> In light of the above, we believe that all substantive issues raised during the review process have been fully addressed, and we kindly ask the Area Chair to take this into account when making the final decision.

---

### Author Response · Authors · 2025-12-03
**Acknowledgments**

We sincerely thank the area chair for their generous time and dedicated efforts, and all reviewers for their valuable participation and constructive feedback.

---

### Meta-Review · Area_Chair_t3sp · 2026-01-09

**Summary:**

Overall, the authors have addressed properly the major concerns from the reviewers. The initial scores are mostly negative (4, 4, 2, and 6). However, after reading carefully the authors' responses, I believe most critical concerns have been addressed properly. The paper is good enough to be considered for publication. Provided below are a summary of the major concerns from each reviewer and my AC comments [AC]. The other comments are mostly for clarification.

Reviewer 72TK (4: marginally below the acceptance threshold. But would not mind if paper is accepted; 5: You are absolutely certain about your assessment.)

o　The paper compares only with MambaIC from CVPR 2025, ignoring LALIC and DCAE. Considering that LALIC and DCAE released code several months ago, a comparison with these methods is essential.

[AC: The additional comparison with LALIC and DCAE during the rebuttal period clearly shows the benefits of the proposed method.]

o　The proposed Learnable Prompt Dictionary is essentially the Attentive State Space Module from mambairv2. The authors do not mention this in the paper, but since they cite mambairv2, they must be aware of it. I believe this is problematic and undermines the paper’s contribution and originality.

[AC: Although the proposed CMIC differs from MambaIRv2 in several significant ways, the additional comparison results indicate that the proposed CMIC offers limited gain (1% BD-rate saving) as compared to the idea from MambaIRv2.]

Reviewer GLne (4: marginally below the acceptance threshold. But would not mind if paper is accepted; 5: You are absolutely certain about your assessment)

o	The main contribution of this paper is to improve the Mamba module in the Mamba-based learned image codec. However, it seems that there are many other differences compared to the previous works, e.g., MambaVC and MambaIC, including the entropy models, etc. The paper lacks detailed performance analysis.

[AC: The additional results with an advanced entropy model confirms the effectiveness of the proposed method.]

o	The paper lacks a performance comparison with the latest methods listed, e.g., [3].

[AC: [3] was related on 25 July 2025 and is a concurrent work. Its focus is on improving the entropy coding model while this work is on improving the transform part. These design aspects are kind of orthogonal. Additional results also confirm the superiority of the proposed method.]

[AC: The reviewer read the rebuttal responses and decided to increase the rating.]

Reviewer QkwG (2: reject, not good enough. But would not mind if paper is accepted; 4: You are confident in your assessment.)

o	It is necessary to clarify the specific differences between the proposed method and Mambairv2 [1].

[AC: This same issue was raised by the other reviewers and has been addressed.]

o	It is necessary to clarify the specific differences between the proposed method and Mambairv2 [1].

[AC: Additional comparison results confirm that CMIC performs comparably to MLICv2 but at the cost of lower complexity. Moreover, the two works address two orthogonal aspects and are shown to be complementary to each other.]

Reviewer ysic (6: marginally above the acceptance threshold. But would not mind if paper is rejected; 4: You are confident in your assessment.)

[AC: the reviewer confirmed that his concerns have been addressed properly and would keep to his positive rating.]

**Reviewer Concerns:**

See the summary section.

**Reviewer Scores:**

The reviewers with scores 4 would increase their ratings. The reviewer with score 2 (reject) has only 2 comments, both of which have been addressed, may increase the rating by 1. The reviewer with score 6 had indicated that he would stick to his positive score.

---

### Decision · Program_Chairs · 2026-01-26

Accept (Poster)